# Optogenetic engineering of STING signaling allows remote immunomodulation to enhance cancer immunotherapy

Yaling Dou[1,3], Rui Chen[1,3], Siyao Liu[1], Yi-Tsang Lee ⬡[1], Ji Jing[1], Xiaoxuan Liu[1], Yuepeng Ke[1], Rui Wang[1], Yubin Zhou ⬡[1,2] ✉ & Yun Huang ⬡[1,2] ✉

The cGAS-STING signaling pathway has emerged as a promising target for immunotherapy development. Here, we introduce a light-sensitive optogenetic device for control of the cGAS/STING signaling to conditionally modulate innate immunity, called 'light-inducible SMOC-like repeats' (LiSmore). We demonstrate that photo-activated LiSmore boosts dendritic cell (DC) maturation and antigen presentation with high spatiotemporal precision. This non-invasive approach photo-sensitizes cytotoxic T lymphocytes to engage tumor antigens, leading to a sustained antitumor immune response. When combined with an immune checkpoint blocker (ICB), LiSmore improves antitumor efficacy in an immunosuppressive lung cancer model that is otherwise unresponsive to conventional ICB treatment. Additionally, LiSmore exhibits an abscopal effect by effectively suppressing tumor growth in a distal site in a bilateral mouse model of melanoma. Collectively, our findings establish the potential of targeted optogenetic activation of the STING signaling pathway for remote immunomodulation in mice.

The cyclic GMP-AMP synthase (cGAS)-stimulator of the interferon gene (STING) pathway plays a crucial role in detecting cytosolic microbial DNA and initiating innate immune defense against invading pathogens[1,2]. Upon sensing the presence of double-stranded DNA, cGAS catalyzes the synthesis of cyclic dinucleotides, such as cyclic guanosine monophosphate-adenosine monophosphate (cGAMP). Acting as a second messenger, cGAMP binds to the transmembrane adaptor protein STING located in the endoplasmic reticulum (ER), thereby activating STING to form oligomers with subsequent translocation to the intermediate compartments between ER and the Golgi complex[3–5]. Oligomeric STING further recruits TANK-binding kinase 1 (TBK1) and the transcription factor interferon regulatory factor 3 (IRF3) to form supramolecular organizing centers (SMOCs)[6]. TBK1-mediated phosphorylation of IRF3 promotes the multimerization of IRF3 and its translocation into the nucleus, where it cooperates with NF-κB to induce the transcription of interferon-beta (*IFNB*), triggering the release of type I interferons

and proinflammatory cytokines to initiate innate immune response[7].

The cGAS-STING pathway has been actively pursued as a therapeutic target given its intimate involvement in antitumor immunity[8,9]. STING agonists have been shown to stimulate the maturation of dendritic cells (DCs) and enhance their ability to present antigens. This ultimately leads to improved priming of T cells and enhances their cytotoxicity against tumor cells[10]. Furthermore, STING activation in macrophages has been shown to enhance their phagocytotic activity toward cancer cells[11], and reprogram macrophages from a pro-tumorigenic M2-like phenotype to a tumoricidal M1-like phenotype[12]. Several natural and synthetic STING agonists have been evaluated in pre-clinical and clinical studies across various tumor models, revealing their tumor-suppressive effects and ability to enhance antitumor immunity[10,12–16]. However, the use of STING agonists raises a significant concern regarding the potential induction of systemic inflammation due to their pleiotropic effects on various immune cell types.

[1]Institute of Biosciences and Technology, Texas A&M University, Houston, TX, USA. [2]Department of Translational Medical Sciences, School of Medicine, Texas A&M University, Houston, TX 77030, USA. [3]These authors contributed equally: Yaling Dou, Rui Chen. ✉e-mail: yubinzhou@tamu.edu; yun.huang@tamu.edu

Furthermore, achieving effective concentration and retention within tumors often necessitates high dosages, which can exacerbate side effects[9]. Conversely, genetically engineering STING modulators within specific immune cells holds promise for enhancing spatiotemporal control and cell-type specificity, thus minimizing side effects associated with STING agonist treatment.

Contrary to its recognized tumor-suppressive role, recent studies have suggested the involvement of the STING pathway in promoting tumor burden and contributing to poorer disease outcomes in murine tumor models[17,18]. While transient activation of the pathway appears to favor tumor suppression, the lasting STING activation can result in chronic inflammation, creating an immunosuppressive tumor environment to promote tumor growth[2,19]. In parallel, STING agonists have the potential to induce the expression of inhibitory molecules, such as programmed death-ligand 1 (PD-L1) and indoleamine 2, 3-dioxygenase 1 (IDO1), which counteract the tumor-suppressive effects[2,20]. Moreover, excessive activation of the STING pathway has been shown to promote apoptosis in T and B cells[21,22]. Additionally, as STING is expressed ubiquitously in multiple cell types, the administration of STING agonists may lead to systemic inflammation, including the development of inflammatory and autoimmune responses[2]. These findings underscore the critical need to develop a method that would allow precise temporal and spatial control over the STING signaling, particularly in antigen-presenting cells such as dendritic cells (DCs), in order to mitigate the aforementioned adverse effects.

Chemically inducible dimerization (CID) is a chemogenetic approach developed for tunable control of protein-protein interactions. However, CID systems lack strict spatial precision and often suffer from irreversibility[23]. In contrast, optogenetics offers tremendous potential for achieving precise spatial and temporal control over physiological processes in live cells and tissues[24–28]. A notable application of this technique is our recent development of an optogenetic approach that enables spatiotemporal control of chimeric antigen receptor (CAR) T-cell-mediated cytotoxicity against tumor cells[29,30]. This approach effectively reduces side effects, such as the cytokine release syndrome and "on-target, off-tumor" cytotoxicity, commonly observed in patients undergoing CAR-T cell therapy.

In this study, we present the development of LiSmore (light-inducible SMOC-like repeats) as an ultra-light-sensitive optogenetic tool based on STING. LiSmore enables spatiotemporal control of STING signaling in living animals. Through non-invasive modulation of STING using blue light, we demonstrate the effective promotion of DC maturation and antigen presentation, resulting in potent sensitization of T cells for efficient cancer cell killing. Furthermore, the combination of LiSmore with immune checkpoint blockade (ICB) synergistically enhances the antitumor efficacy in an immunosuppressive lung cancer model that remains unresponsive to ICB treatment alone. Remarkably, through photo-activation at its primary administration site, LiSmore demonstrates the ability to exert a tumor-suppressive effect on distal tumors in a bilateral melanoma model, showcasing its capability to trigger a desirable abscopal effect. These findings establish the feasibility of utilizing wireless optogenetic immunomodulation in vivo for cancer intervention, highlighting the potential of LiSmore in advancing precision immunotherapy.

## Results

### Design of LiSmore for optogenetic activation of STING signaling

STING consists of an N-terminal domain encompassing multiple transmembrane segments (amino acid, aa 1–153), a dimerization and ligand-binding domain (aa 154–339), and a C-terminal tail (CTT, aa 340–379)[31,32]. Upon binding to cGAMP, STING undergoes oligomerization, accompanied by the release/exposure of the CTT. The CTT domain acts as a scaffold for heterotypic interactions with TBK1 and IRF3[32]. Specifically, the clustered STING-CTT recruits the downstream TBK1 via a conserved PLPLRT/SD motif (aa 371–379)[32]. The binding of TBK1 to this motif promotes phosphorylation of S366 in STING-CTT, which subsequently recruits IRF3 to the pLxIS motif (aa 362–366)[31–33]. The binding of TBK1 and IRF3 to the CTT licenses IRF3 phosphorylation by TBK1 and ultimately activates the type I interferon pathway[32] (Fig. 1a). Inspired by this activation mechanism, we engineered an optogenetic device that mimics STING signaling using light. To achieve this, we fused the CTT domain to an optical multimerizer, the N-terminal photolyase-homologous region of *Arabidopsis* cryptochrome 2 (CRY2), which undergoes monomer-to-oligomer transition following blue light illumination[23,34,35] (Fig. 1a). We hypothesized that blue-light stimulation would induce the clustering of CRY2-CTT, hence mimicking agonist-induced STING activation and subsequently activating the type I IFN pathways. In our initial design, we fused one or two copies of the truncated CTT (aa 355–379 containing the TBK1 and IRF3 binding motifs) or the entire CTT (aa 341–379) to CRY2 (Fig. 1b). When expressed in HEK293T cells, only the construct with tandem repeats of CTT (2×CTT) showed upregulation of *IFNβ* expression upon photostimulation (Fig. 1b and Supplementary Fig. 1b). Furthermore, the tandem repeats of CTT should be placed immediately after CRY2, as the CRY2-mCh-2×CTT construct failed to effectively activate *IFNβ* expression upon light exposure (Fig. 1b and Supplementary Fig. 1a). Therefore, we chose mCh-CRY2 fused with two copies of CTT (2×CTT; hereafter referred to as CRY2-pLxIS) as our lead design for further characterization and optimization.

Given the reported STING-TBK1 interaction upon STING activation[6], we examined whether mCh-CRY2-pLxIS and TBK1-YFP could co-localize when co-expressed in mammalian cells before and after light stimulation. In the absence of blue light, both mCh-CRY2-pLxIS and TBK1-YFP exhibited an even distribution throughout the cytosol of HeLa cells (Fig. 1c). Upon blue light illumination, mCh-CRY2-pLxIS rapidly formed puncta ($t_{1/2}$ = 92 ± 6 s), accompanied by co-clustering of TBK1-YFP (Fig. 1c and Supplementary Movie 1). By contrast, the control construct mCh-CRY2 did not show appreciable co-localization with TBK1-YFP after light stimulation (Supplementary Fig. 1b and Supplementary Movie 2). To further confirm the functional consequence of co-clustering, we transduced the THP-1 monocyte cell line with a lentivirus encoding mCh-CRY2-pLxIS. In these transduced THP-1 cells, we observed increased levels of phosphorylated TBK1 (p-TBK1) and phosphorylated IRF3 (p-IRF3) following 0.5 or 1-h photostimulation (Fig. 1d). These findings indicate that upon blue light stimulation, mCh-CRY2-pLxIS interacts with TBK1 to induce IRF3 phosphorylation.

To further investigate the activation of downstream target genes, we expressed either mCh-CRY2-pLxIS or mCh-CRY2-Ctrl in HEK293T cells and performed quantitative real-time PCR (qRT-PCR) analysis of signature genes representative of STING activation. Upon blue light illumination, the CRY2-pLxIS group exhibited a significant increase (8-40 fold) in the STING pathway-related signature genes, including the radical S-adenosyl methionine domain containing 2 (*RSAD2*), C-X-C motif chemokine ligand 10 (*CXCL10*), interferon beta (*IFNB*), interferon-induced protein with tetratricopeptide repeats 1/2/3 (*IFIT1*, *IFIT2*, *IFIT3*), and interferon-stimulated 15-KDa protein (*ISG15*), compared to the CRY2-Control (CRY2-Ctrl) group and the dark group (Fig. 1e). At the protein level, mCh-CRY2-pLxIS induced the secretion of downstream IFNβ upon light stimulation, while mCh-CRY2 did not (Fig. 1f). Importantly, since HEK293T cells have minimal endogenous STING expression, these results suggest that mCh-CRY2-pLxIS does not require endogenous STING for its activity. Further supporting this conclusion, we expressed mCh-CRY2-pLxIS in a murine macrophage cell line J774A.1, which also led to the upregulation of *Rsad2*, *Cxcl10*, *Ifnb* and nitric oxide synthase 2 (*Nos2*) upon light stimulation (Supplementary Fig. 2a), as well as the secretion of IFNβ after light stimulation (Supplementary Fig. 2b). In contrast, J774A.1 cells expressing mCh-CRY2 as a control did not exhibit significant IFNβ secretion (Supplementary Fig. 2b). Additionally, the activation markers CD80

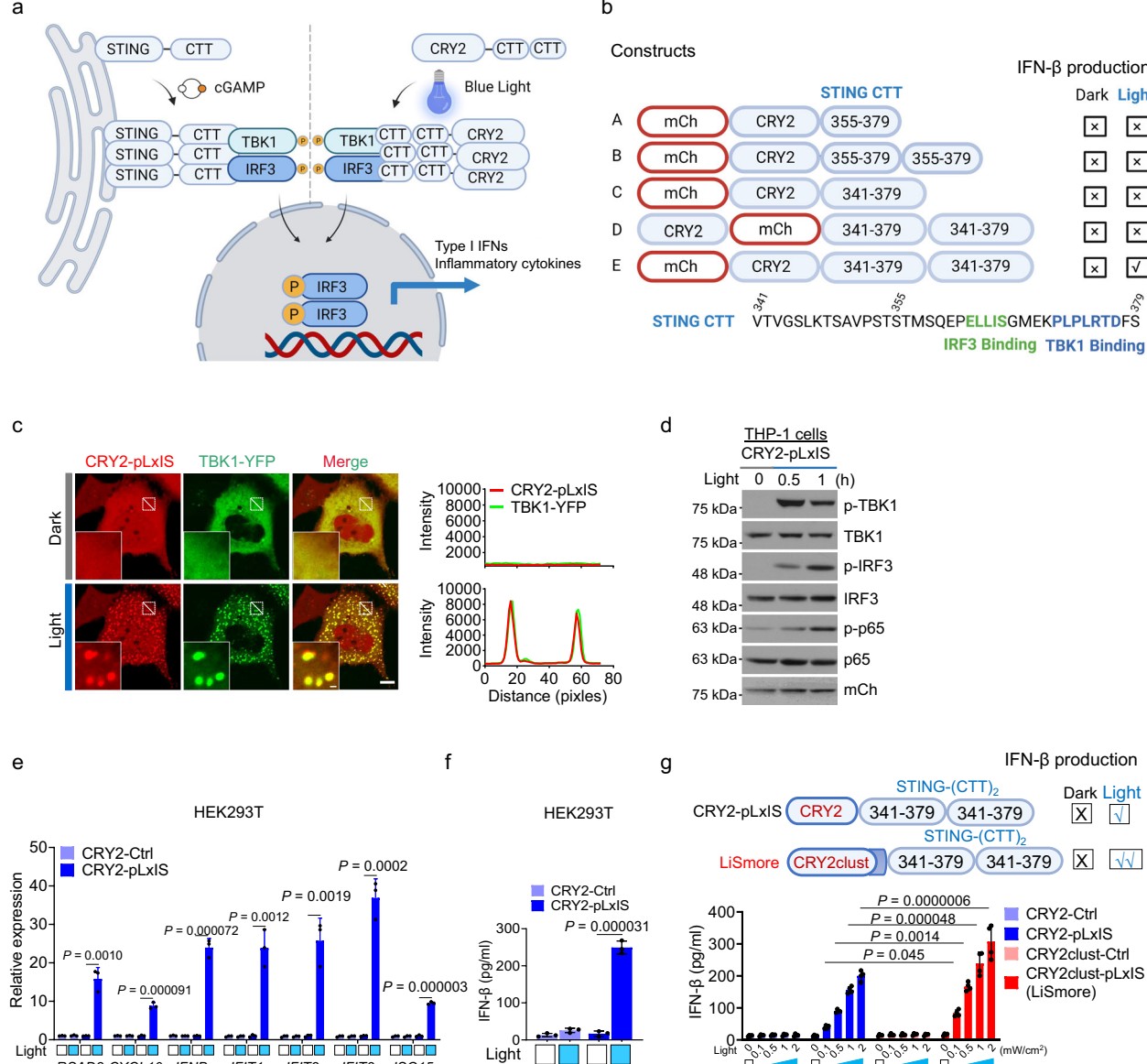

**Fig. 1 | Optogenetic modulation of the STING pathway using LiSmore.**
**a** Schematic illustrating the comparison between physiological activation of the STING pathway upon cGAMP stimulation (left) and light-induced STING pathway activation using LiSmore (right). Photostimulation leads to CRY2-driven multi-merization of the STING C-terminal tail (CTT), resulting in downstream TBK1 and IRF3 recruitment and phosphorylation. Phosphorylated IRF3 forms a dimer, translocates into the nucleus, and activates type I interferon responses. **b** Domain architectures of CRY2-pLxIS and its variants. The photolyase-homology region of CRY2 is fused with one or two copies of STING-CTT. The TBK1 and IRF3 docking sequences are shown in blue and green, respectively. **c** Confocal images of HeLa cells co-expressing mCh-CRY2-pLxIS (red) and TBK1-YFP (green) before and after photostimulation. The intensity profiles of TBK1-YFP (green) and CRY2-pLxIS (red; across the white line) in response to photostimulation are plotted on the right. Scale bar, 10 μm (1 μm in the magnification). Data are representative of three independent experiments. See Supplementary Movie 1. **d** Immunoblot analysis of TBK1, IRF3, and p65, as well as their phosphorylated forms, in THP-1 cells

expressing mCh-CRY2-pLxIS before and after photostimulation at the indicated time points. Data are representative of three independent experiments. **e, f** Quantification of mRNA expression levels of the STING pathway-related signature genes by qRT-PCR, including *RSAD2* (radical S-adenosyl methionine domain containing 2), *CXCL10* (C-X-C motif chemokine ligand 10), *IFNB* (interferon beta 1), *IFIT1-3* (interferon-induced protein with tetratricopeptide repeats 1, 2, and 3) and *ISG15* (interferon-stimulated 15 kDa protein) (**e**) and secreted IFNβ with ELISA (**f**) in sorted mCherry⁺ HEK293T cells transfected with CRY2-pLxIS and CRY2-control. Cells were exposed to pulsed blue light (470 nm, 4 mW/cm², 30 s ON/OFF cycle for 8 h). $n = 3$ biological replicates; mean ± SD; Two-sided unpaired Student's t-test. **g** Determination of secreted IFNβ using ELISA at varying light intensities. FACS-sorted HEK293T cells expressing the indicated constructs (mCherry⁺) were either kept in the dark or exposed to pulsed blue light delivered at 0.1, 0.5, 1, and 2 mW/cm² (30 s ON and 30 s OFF for 8 h). $n = 4$ biological replicates (mean ± SD); Two-sided unpaired Student's t-test.

and CD86 were upregulated in the mCh-CRY2-pLxIS group following blue light illumination (Supplementary Fig. 2c).

The limited tissue-penetrating efficiency of blue light-activatable optogenetic tools poses a challenge for their applications in living organisms[36–38]. To overcome this limitation and explore the possibility of controlling STING signaling in vivo simply using blue light without

invasive fiber optics or upconversion nanoparticles as the NIR-to-blue light transducer, we explored the use of an ultra-light sensitive version of CRY2, CRY2clust. This variant extends the C-terminal amino-acid sequence of CRY2 with nine additional residues (ARDPPDLDN), enabling optogenetic stimulation in deep brain regions of mice with non-invasive light illumination at very low light density (0.1 mW/

cm²)[35,39]. When exposed to blue light, CRY2clust undergoes rapid cluster formation within seconds, and these clusters can be disassembled upon withdrawal of photostimulation[35]. Indeed, we observed the reversible clustering of the hybrid protein and repeated recruitment of TBK1 in response to ON/OFF cycles of blue light stimulation in transfected HEK293T cells (Supplementary Fig. 3 and Supplementary Movie 3), providing strong evidence for the excellent reversibility of this optogenetic tool.

Subsequently, we conducted a comparative analysis to assess the performance of our optimized CRY2clust-pLxIS tool and the prototypic CRY2-pLxIS in HEK293T cells. Both groups were exposed to varying power densities (0.1, 0.5, 1, or 2 mW/cm²) of blue light pulses (470 nm; 30 s ON and 30 s OFF cycles for 8 h). We found that the CRY2clust-pLxIS design consistently outperformed the original CRY2-pLxIS construct under all conditions (Fig. 1g). Notably, CRY2clust-pLxIS remained responsive even at the lowest tested light intensity of 0.1 mW/cm², which exhibited an approximately 8-fold increase in IFNβ secretion compared to the performance of CRY2clust-pLxIS. In contrast, CRY2-pLxIS displayed weak or barely detectable response to light stimulation at 0.1 and 0.5 mW/cm². In aggregate, we have successfully recapitulated the STING→p-TBK1→pIRF3→IFN-I signaling cascade using a compact single-component optogenetic tool (with a size of <2 kb) that is compatible with existing viral packaging systems, allowing wireless regulation of STING activation using simple light pulses. We named this ultra-photosensitive CRY2clust-pLxIS construct as "LiSmore", which stands for light-inducible SMOC-like repeats.

## LiSmore drives type I interferon (IFN-I) production through STING signaling

The cGAS-STING signaling, which drives TBK1/IRF3 signaling-dependent type I IFN production in response to tumor-derived DNA, plays an indispensable role in tumor surveillance mediated by antigen-presenting cells, such as DCs and macrophages[40]. To investigate the potential application of LiSmore in cancer intervention, we sought to combine our optogenetics tool with DC-based immunotherapy. Following the oligomerization of STING induced by cGAMP, the activation of TBK1/IRF3 signaling ensues, resulting in the release of type I interferons (IFN-I) such as IFN-β[41]. To assess the activation of type I interferon production mediated by LiSmore, we first examined type I interferon production in vitro using bone marrow-derived dendritic cells (BMDCs), which are critical for cytotoxic T cells-mediated antitumor function and are commonly employed in cancer treatment with cancer vaccines[42,43]. Following in vitro culture of bone marrow progenitors with granulocyte–macrophage colony-stimulating factor (GM-CSF) for 6 days, BMDCs were transduced with retroviruses encoding either Control (GFP-CRY2clust as the vector control) or LiSmore (GFP-CRY2clust-pLxIs) twice on Day 7 and Day 8 (Fig. 2a). GFP⁺CD11c⁺ DCs were subsequently sorted to a purity of over 90% for further analysis (Fig. S4). We first measured type I interferon secretion by ELISA in purified GFP⁺ BMDCs, in the absence of the presence of pulsed blue light illumination. When exposed to blue light, LiSmore significantly increased the production of IFN-α and IFN-β production compared to the control group or the dark groups (Fig. 2b). Furthermore, photo-activated LiSmore led to enhanced production of other

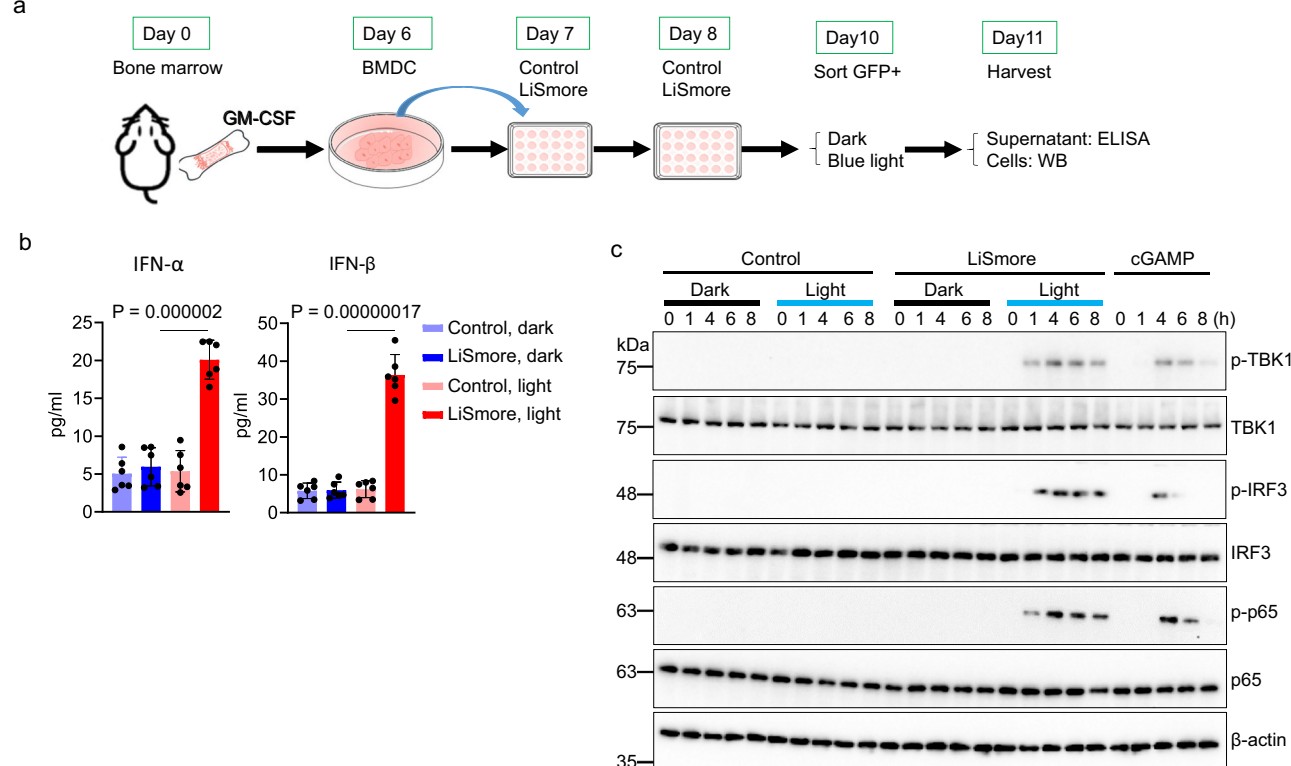

**Fig. 2 | Photo-activated LiSmore drives type I interferon production in BMDCs.** **a** Scheme for the experimental setup. Bone marrows from C57BL/6J mice were cultured in GM-CSF to induce BMDCs. On days 7 and 8, BMDCs were transduced twice with Control (GFP-CRY2clust) or LiSmore (GFP-CRY2clust-pLxIS). After 36 h, GFP⁺ BMDCs were sorted and replanted in 48-well plates. BMDCs were either shielded from light (Dark) or exposed to blue light (Light; 470 nm, 1 mW/cm²; 20 s ON, 5 min OFF cycles). **b** ELISA analysis of secreted IFN-α and IFN-β concentrations in the supernatants collected from the four indicated groups with or without light stimulation for 18 h. $n = 6$ biological replicates (mean ± SD); Two-sided unpaired Student's $t$-test. **c** Immunoblot analysis of TBK1, IRF3, and p65 phosphorylation in BMDCs in the absence (black bar) or presence of light stimulation (blue bar; 470 nm, 1 mW/cm²; 20 s ON, 5 min OFF cycles) for the indicated durations (0–8 h). Non-transduced BMDCs were treated with 2'3'-cGAMP (2 µg/ml) under the same timeframe side-by-side. Results are representative of two independent experiments.

proinflammatory cytokines, including CXCL10, CXCL1, GM-CSF, IL-6, IL-1β, IL-12p70, and TNFα (Supplementary Fig. 5).

To gain a mechanistic understanding of LiSmore-mediated effect on dendritic cells, we next examined the TBK1/IRF3/P65 phosphorylation levels in LiSmore-transduced BMDCs by immunoblotting. For comparison, we included CRY2clust as a negative control and cGAMP treatment as a positive control. Interestingly, we observed distinct activation profiles of the downstream effectors within the STING pathway when comparing LiSmore and cGAMP treatment. In the cGAMP-treated group, we noticed a relatively delayed onset of TBK1/IRF3/P65 phosphorylation around 4 h after cGAMP treatment. Moreover, this phosphorylation substantially declined at 6 and/or 8 h post-treatment, indicating a relatively short-term activation of the STING pathway in response to cGAMP stimulation. In contrast, LiSmore showed an earlier and more sustained STING activation profile, appearing within 1 h post-photostimulation and persisting over the course of 8 h (Fig. 2c). Collectively, these findings demonstrate that LiSmore, as an engineered innate immunity actuator, is capable of eliciting rapid and prolonged activation of the innate immune response through the STING pathway. This further highlights the unique and advantageous properties of LiSmore in modulating innate immunity for potential therapeutic intervention.

## LiSmore enhances both BMDC and T-cell activities

One of the key challenges facing DC vaccination-based immunotherapy is to maintain the maturation status of dendritic cells and enhance their antigen presentation[44]. To assess the effect of LiSmore on the expression of surface molecules of mature DCs that are important for the activation of CD8 cytotoxic lymphocytes (CTLs), we measured hallmark markers indicative of dendritic cell maturation using flow cytometry. These markers include major histocompatibility complex class I (MHC-I) and class II (MHC-II), co-stimulatory molecules CD80, CD86, or CD40, as well as chemokine receptor CCR7 that plays a vital role in DCs migration to draining lymph nodes for T-cell engagement[45]. As depicted in Fig. 3a, we transduced bone marrow-derived DCs (BMDCs) with retroviruses expressing either the Control or LiSmore construct. Subsequently, we divided the transduced cells into two groups: the dark group, where cells were shielded from light exposure, and the light group, where cells were exposed to pulsed blue light illumination for 18 h. As a positive control for comparison, untransduced BMDCs (the blank group) were treated with a STING agonist, 2′3′-cGAMP, under the same condition. In the absence of light, both the control and LiSmore DCs displayed marginal or low expression levels of MHC-I/II (H2kb or IA/IE), CD80, CD86, CD40, and CCR7. However, a significant increase in the surface expression of all six molecules was observed following photostimulation (Fig. 3b, c). cGAMP treatment also enhanced the expression of MHC-I/II, CD80, CD86, and CD40. However, the increase in CCR7 expression induced by cGAMP was not as prominent as observed for the LiSmore-light group (red versus black bars; Fig. 3b, c).

To investigate the effect of LiSmore on T cells ex vivo, we isolated CD8+ T cells from OT-1 mice, which possess a T-cell receptor specific for the ovalbumin-derived peptide SIINFEKL (OVAp). Isolated T cells were then mixed with LiSmore-BMDCs that had been pre-pulsed with OVAp for functional analysis (Fig. 3d). To track and monitor the proliferation of CD8+ T cells, we utilized CellTrace Violet (CTV) staining to label them prior to co-culture with OVAp-pulsed BMDCs. In the presence of blue light stimulation, CD8+ T cells expressing LiSmore exhibited increased cell division and proliferation compared to the control group shielded from light. Moreover, the LiSmore group displayed a higher division index compared to the group treated with cGAMP (Fig. 3e), indicating a more robust proliferation of T cells. Next, we aimed to evaluate the ability of LiSmore to promote T-cell priming by co-culturing OT-1 CD8+ T cells with OVAp pre-pulsed LiSmore-BMDCs at a ratio of 2:1 (Fig. 3d). The activation of OT-1 CD8+ T cells was assessed by measuring IFN-γ production at 18 h post-coculture. Flow cytometry analysis revealed that LiSmore-BMDCs strongly stimulated the activation of OT-1 CD8 T cells upon exposure to blue light, as evidenced by a higher level of IFN-γ production compared to the cGAMP group (Fig. 3f).

To validate the ability of LiSmore to enhance DC-mediated antigen sensing and cross-presentation of tumor antigens for T cell priming, we used B16-OVA melanoma cells stably expressing chicken ovalbumin (OVA) as a model antigen (Fig. 3g). These cells were co-cultured with LiSmore-BMDCs in the absence or presence of blue light (Fig. 3h, i). Flow cytometry analysis revealed that BMDCs expressing LiSmore displayed an approximately three-fold higher surface presentation of the OVAp/MHC-I complex (SIINFEKL–MHC-I) under blue light illumination, even surpassing the increase observed in the group treated with cGAMP (Fig. 3h).

We next moved on to evaluate the capacity of LiSmore-BMDCs to stimulate cytotoxic T lymphocyte (CTL) response against B16-OVA melanoma cells in a ternary co-culture assay (Fig. 3i). We assessed CTL-mediated cytotoxicity by measuring the release of lactate dehydrogenase (LDH) in the cell culture medium. Among all the groups, LiSmore-BMDCs exhibited the highest level of LDH release (52%) in response to blue light, which was approximately two-fold stronger than BMDCs treated with cGAMP (Fig. 3i). In contrast, the control groups showed minimal or low LDH release. To validate the specificity of T cell-mediated cytotoxicity toward tumor cells, B16-OVA melanoma cells were pre-stained with CellTrace Violet prior to co-culturing with BMDCs and CD8+ T cells. Flow cytometry analysis was then conducted to assess B16-OVA cell death, indicated by double-positive staining of the violet dye and 7-AAD (Supplementary Fig. 6a). Upon blue light illumination, the LiSmore group demonstrated enhanced T cell-mediated cytotoxicity towards B16-OVA cells. The extent of B16-OVA cell death (Violet+7-AAD+) was approximately 2-4 fold higher in the LiSmore group upon photostimulation compared to either the cGAMP group or the control groups (Supplementary Fig. 6b, c). These compelling findings establish that LiSmore efficiently promotes the activation of antigen-presenting cells and enhances cross-presentation of tumor antigens, effectively priming effector CD8+ T cells for targeted tumor killing in a light-dependent manner.

## LiSmore promotes CTL response to mitigate melanoma burden

LiSmore is built upon an ultra-photosensitive CRY2clust photoswitch, which has been previously applied for transcranial control of neuronal activities in deep brain regions of mice[39]. We therefore reasoned that LiSmore could likewise drive STING-dependent molecular events in tissues beneath mouse skin through non-invasive light illumination. To test this idea, we designed a light-illuminating cage with an LED light source capable of emitting low-intensity blue light at 470 nm (~2 mW/cm$^2$; Fig. 4a). We first assessed the functionality of this platform by examining in vivo STING activation in dendritic cells isolated from tumor-draining lymph nodes (tdLNs). We directed the LED light source toward the skin above the inguinal lymph nodes in the mouse hind leg (red circle; Supplementary Fig. 7a, b). We tested three different blue light intensities (1, 2 and 4 mW) and compared the performance of LiSmore with the prototypic CRY2-pLxIS construct in a mouse melanoma model (Supplementary Fig. 7c). Utilizing the immunostaining level of p-TBK1 as a convenient readout, we observed that LiSmore-DCs (indicated by GFP+CD45.2+) from tumor-draining lymph nodes displayed a notable increase in p-TBK1 staining following blue light stimulation (Fig. S7d). Notably, even at a low light density (1 mW/cm$^2$), LiSmore exhibited in vivo activation, as evidenced by a pronounced increase in p-TBK1 fluorescence. In contrast, the prototype CRY2-pLxIS group displayed no discernible response to the 1–4 mW blue light stimulation (Supplementary Fig. 7d). Collectively, these results provide strong evidence that LiSmore-equipped immune cells in the inguinal

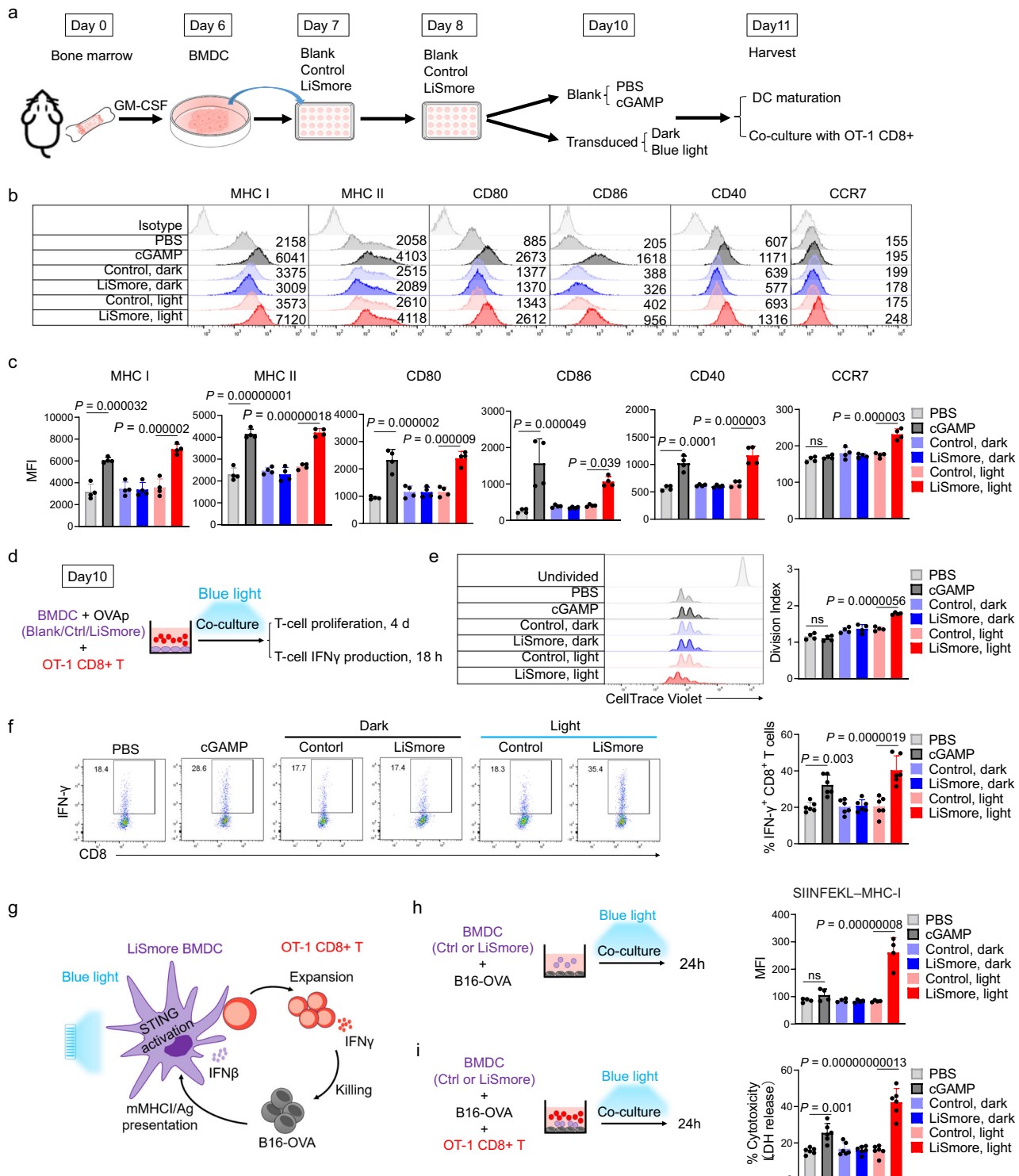

lymph nodes retain their capacity to respond to non-invasively delivered blue light owing to their exceptional light sensitivity.

Considering the intimate involvement of the cGAS-STING pathway in antitumor immunity[7,46], we conducted further investigations to evaluate the in vivo antitumor efficacy of LiSmore using a murine model of melanoma. To initiate tumor growth, we subcutaneously injected B16-OVA melanoma cells into wild-type (WT) CD45.1 B6 recipient mice (Fig. 4b). On day 4 post-inoculation, we sorted GFP+ LiSmore dendritic cells (LiSmore-DCs) that were previously pulsed with ovalbumin and transferred them into the melanoma-bearing CD45.1 B6 recipient mice. On the following day, we adoptively transferred

CD45.2+CD8+OT-I T cells into the melanoma-bearing CD45.1 recipient mice via retro-orbital injection. In this well-established tumor model, adoptively transferred OT-1 CD8 T cells specifically recognize the OVAp epitope presented by dendritic cells[47]. However, these T cells alone are insufficient to effectively suppress tumor growth unless combined with other types of immunotherapeutic strategies such as DC-based immunomodulation[48,49]. To assess the potential antitumor effect of LiSmore, we exposed the mice to pulsed blue light for a duration of 7 days (470 nm; 2 mW/cm²; 30 min ON/OFF cycle for 6 h per day). Throughout this period, we monitored tumor growth at 2-day intervals (Fig. 4a, b) by using phosphate-buffered saline (PBS) and

**Fig. 3 | LiSmore enables photo-inducible maturation and antigen presentation in BMDCs. a** Experimental setup. Bone marrow cells from C57BL/6J mice were cultured in GM-CSF to induce BMDCs. On days 7 and 8, BMDCs were infected with Control, LiSmore, or left untreated (blank). After 36 h, transduced BMDCs were either shielded (dark) or exposed to pulsed blue light (light; 470 nm, 1 mW/cm²; 20 s ON, 5 min OFF) for 18 h. Blank BMDCs were stimulated with cGAMP and PBS. **b, c** Flow cytometry profiles (**b**) and mean fluorescent intensity (MFI) quantification (**c**) of H2Kb or MHCI, IA/IE, MHCII, CD86, CD80, CD40) and CCR7 in the indicated BMDCs. Plots were gated on viable CD11c⁺GFP⁺ events. *n* = 4 independent biological replicates (mean ± S.D.); one-way ANOVA. **d** Experimental setup for ex vivo OT-1 CD8⁺ T cell activation. OVAp pulsed control or LiSmore BMDCs were co-cultured with OT-I CD8⁺ T cells. Cells were exposed to pulsed blue light for 18 h (470 nm, 20 s ON, 5 min OFF, 1 mW/cm²). Untransduced BMDCs/OVAp were also co-cultured with OT-1 CD8⁺ T cells and treated with 2′3′-cGAMP (2 µg/ml) or PBS. **e** Proliferation

(CellTrace Violet, CTV) of OT-1 CD8⁺ T cells on day 4. The division index is shown in the right bar graph (**e**). *n* = 4 independent biological replicates (mean ± S.D.); one-way ANOVA. **f** Left, representative flow cytometry profiles indicative of IFN-γ production from OT-1 CD8⁺ T cells 18 h. Right, quantification of IFN-γ production in OT-1 CD8⁺ T cells. *n* = 6 independent biological replicates (mean ± S.D.); one-way ANOVA test. **g** Experimental design for assessing cross-presentation and CTL-mediated cytotoxicity. OT-1 CD8⁺ T cells, LiSmore-BMDCs, and B16-OVAp cells were mixed at a 2:1:1 ratio with or without light illumination (470 nm, 1 mW/cm²; 20 s ON, 5 min OFF) for 24 h. **h** Quantification of OVAp presentation on SIINFEKL−MHC-I in LiSmore-BMDCs or control BMDCs with and without photostimulation. *n* = 4 independent biological replicates (mean ± S.D.); one-way ANOVA. **i** LDH release assay to evaluate OVA-specific CD8⁺ T cell killing. *n* = 6 independent biological replicates from three experiments (mean ± SD); one-way ANOVA.

cGAMP (10 µg via intramuscular injection with three dosages)[50] as negative and positive controls, respectively. As anticipated, the application of photostimulation to the LiSmore group led to a pronounced inhibition of tumor growth, with the tumor volume reduced to a level comparable to that of the cGAMP group (Fig. 4c, d and Supplementary Fig. 8). Together, these findings highlight the robust light-dependent tumor-suppressive effects exhibited by LiSmore-DCs, effectively impeding tumor growth in living mammals.

Given the critical role of dendritic cell-mediated cross-presentation in the priming and activating naïve tumor-specific CD8⁺ T cells, we proceeded to evaluate the impact of LiSmore on antigen presentation, as well as the activation and proliferation of CD8⁺ T cells. We observed that GFP⁺ LiSmore-DCs, isolated from tumor-draining lymph nodes, displayed an enhanced capacity for antigen presentation, as indicated by elevated levels of OVAp/MHC-I complex staining. Furthermore, these DCs showed CD80 staining at 18 h after blue light exposure, indicative of sustained dendritic cell maturation (Fig. 4e). Next, we assessed the activation of donor CD45.2⁺CD8⁺ T cells at 4 h following the final treatment in the melanoma-bearing CD45.1 recipient mice. Activation of CD8 T cells was evaluated by measuring the surface expression of the activation marker CD69, the proliferation marker Ki67, and IFN-γ production. When compared to both the control group and the LiSmore group without photostimulation, the LiSmore group displayed 2-4 fold increase in CD69, Ki67, and IFN-γ staining following light treatment in the tumor sites (Fig. 4f). Notably, we observed an increased presence of CD69⁺CD8⁺ T cells within the tumor sites (Fig. 4g), indicating enhanced infiltration of effector T cells (or tumor-infiltrating lymphocytes, TILs) into the tumor microenvironment. Congruently, these findings confirm that LiSmore effectively stimulates cross-presentation of tumor antigens, thereby boosting CD8-mediated antitumor functions in vivo.

## LiSmore enhances anti-PD-L1 treatment efficacy in an immuno-suppressive cancer model

Immune checkpoint blockade (ICB), which targets negative regulators of T cells (e.g., PD-1, PD-L1, and CTLA4), has shown promising therapeutic outcomes in certain cancer types[51,52]. However, some malignancies, including the Lewis lung carcinoma (LL/2), remain largely unresponsive to ICB treatments due to an immunosuppressive tumor microenvironment[53]. These tumors are regarded as "cold" tumors, characterized by a lack of tumor antigens, defects in antigen-presenting cells, and/or absence of T-cell activation[54,55]. Indeed, in our own hand, we found that LL/2 tumors did not respond to anti−PD-L1 therapy (Supplementary Fig. 9). Given the demonstrated ability of LiSmore-DCs to enhance antigen presentation and prime cytotoxic T cells, we set out to test the idea of applying LiSmore to overcome this resistance. We compared the efficacy of LiSmore-DCs with or without anti−PD-L1 treatment in a mouse model of LL/2 tumor (Fig. 5a). The combination of LiSmore-DCs and anti-PD-L1 treatment under blue light stimulation showed the most potent tumor suppression compared to

control mice receiving either Control-DCs or LiSmore-DCs without light stimulation (Fig. 5b–d). When exposed to light, tumor-bearing mice treated with LiSmore-DCs exhibited an appreciable reduction in tumor growth and weight, indicating that LiSmore-BMDCs alone were effective in controlling PD-L1-insensitive tumor burden. Collectively, the combination of LiSmore-DCs with anti-PD-L1 treatment has the potential to synergistically enhance the antitumor efficacy to overcome ICB treatment resistance.

## LiSmore enables precise control of STING activation and elicits abscopal effect in vivo

To illustrate the precise control and potential abscopal effect facilitated by LiSmore, we employed a syngeneic mouse model of melanoma without the ectopic expression of OVA as a surrogate tumor antigen. In this more wild-type like tumor model, we used B16-F10 melanoma cells (without OVA transduction) derived from the skin tissue of a mouse with melanoma[56–58] (Fig. 6a), rather than the B16-OVA cells. In order to assess the in vivo spatiotemporal control of STING activation mediated by LiSmore, we subcutaneously implanted B16-F10 cells into both flanks of CD45.1 B6 mice (Supplementary Fig. 10a). Following tumor establishment at day 7, the right flank (the primary site) of the mice was exposed to a pulsed blue light beam (470 nm; ~2 mW/cm²; 20 s ON + 5 min OFF; 18 h), while the left side (the distal site) was shielded from light using aluminum foil. We found that increased staining of p-TBK1 and p-IRF3 staining was only detected in the primary site subjected to photostimulation, while the distal site shielded from light displayed no appreciable activation (Supplementary Fig. 10b, c). These results provide compelling evidence for the precise spatial and temporal control achieved through the implementation of our optogenetic strategy.

As described above, LiSmore and cGAMP exhibit differential activation of the innate immune response. In particular, LiSmore induces more rapid and sustained activation of TBK1 and IRF3 (Fig. 2c), along with enhanced antigen presentation and cytotoxicity mediated by CTLs (Figs. 3 and 4). These observations prompted us to conduct a side-by-side comparison of the potential abscopal effect and systemic toxicity between these two interventional strategies. For this purpose, we generated bilateral B16-F10 melanoma-bearing mice and examined potential abscopal effect on the distal site (left flank) by exposing the primary site (right flank) to pulsed blue light (Fig. 6a). In parallel, we performed similar experiments with three intratumoral injections of either cGAMP or PBS on the primary sites using the same melanoma model. Both blue light illumination and cGAMP administration led to significant tumor-suppressive effects at the primary sites (Fig. 6b, c). Interestingly, the LiSmore group, but not the control or cGAMP groups, showed a noticeable reduction in tumor sizes at the distal sites (Fig. 6b, c), implying a possible abscopal effect induced by LiSmore upon light stimulation. To compare the induction of antitumor cytokines by LiSmore-DCs and cGAMP, we analyzed the expression of IFN-β and IFN-γ at 4 and 18 h post-treatment (Supplementary Fig. 11a). No

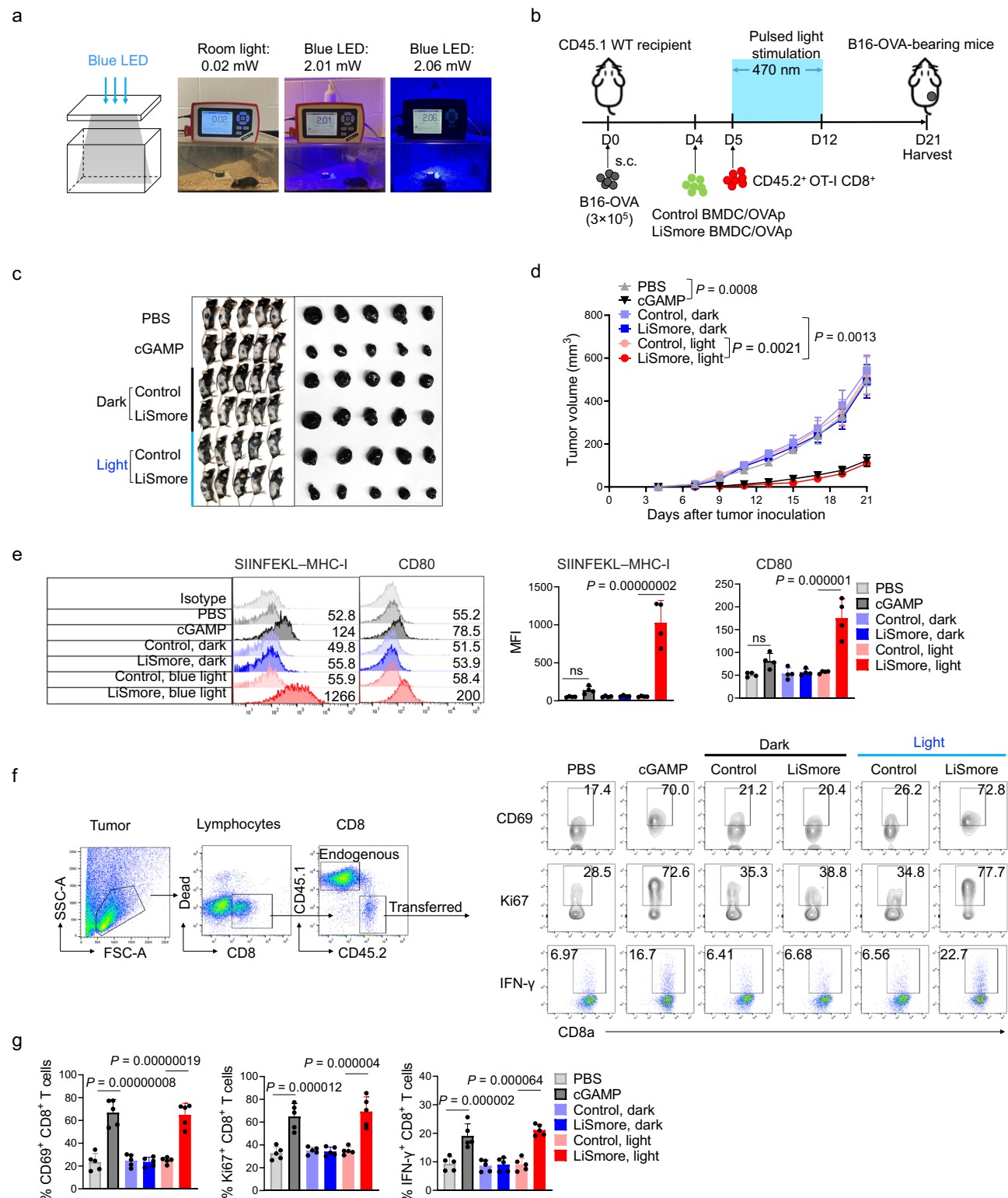

significant difference was observed between the LiSmore and cGAMP groups at 4 h post-treatment. However, at the later time point (18 h), the LiSmore group displayed higher and more sustained production of IFN-β and IFN-γ in both the primary sites and in the sera compared to the cGAMP group (Supplementary Fig. 11b). The difference might partially explain the observed discrepancy in abscopal effect.

Lastly, we compared the potential systemic toxicity associated with LiSmore-DCs and cGAMP in the same mouse model. Consistent with previous reports[9,59], cGAMP treatment resulted in considerable

systemic side effects, as evidenced by elevated levels of alanine transaminase and aspartate transaminase (ALT and AST; indicative of liver function), urea (kidney), and serum IL-6 (Supplementary Fig. 12). In contrast, the systemic administration of LiSmore-DCs did not appear to cause a higher degree of side effects when compared to the PBS or control groups (Supplementary Fig. 12). These findings suggest that, compared to the STING agonist cGAMP, LiSmore is capable of eliciting a more durable innate immune response and triggering more robust antitumor immunity. Consequently, LiSmore effectively

**Fig. 4 | LiSmore enhances the antitumor effect of OT-1 CD8 T cells to reduce melanoma burden in a light-dependent manner. a** Images showing the home-made LED arrays for photostimulation. The blue LED device was installed on the cage lid to enable photoactivation of LiSmore-expressing dendritic cells in the recipient mice. The pictures on the right showed the ambient light density when LED was switched off or turned on. **b** Schematic illustrating the in vivo testing setup using a B16-OVA melanoma model. $3 \times 10^5$ B16-OVA cells were injected (s.c.) in the flank of CD45.1 mice. Mice received PBS, or cGAMP (10 μg/mouse) at days 5, 8, and 11 after tumor inoculation, or were transferred with OVAp-loaded BMDCs expressing Control or LiSmore at day 4, followed by adoptive transfer of CD45.2⁺ OT-1 CD8⁺ T cells at day 5. The mice were then either subjected to photo-stimulation (light) for 7 days (470 nm; ~2 mW/cm²; 30 min ON/OFF cycles for 6 h per day) or shielded from light stimulation (dark). **c** Representative images of melanoma-

bearing mice (left) and isolated tumors (right) for each group on day 21. **d** Quantification of tumor sizes. $n = 5$ mice (mean ± SD). Two-sided unpaired Student's $t$-test. **e** Flow cytometry profiles (left) and quantification of mean fluorescence intensity (MFI) of the OVAp/MHC-1 complex (SIINFEKL–MHC-I; middle) and CD80 (right) on the surface of adoptively transferred DCs isolated from tumor-draining lymph nodes (tdLNs) at 18 h in the absence (dark) or presence (light) of photostimulation. Migrated DCs were defined in tdLNs by gating CD45.2⁺GFP⁺ cells. $n = 4$ independent biological replicates (mean ± SD); one-way ANOVA test. **f** Flow cytometry analysis of CD69, Ki67 and IFN-γ expression in CD45.2⁺ OT-1 CD8⁺ T cells at 4 h after the final treatment. Left, the gating strategy. Right, representative FACS profiles. **g** Quantification of CD69⁺CD8⁺ tumor-infiltrating lymphocytes (TILs; left), Ki67⁺CD8⁺ TILs (middle), and IFNγ⁺CD8⁺ TILs (right) in tumors collected from mice shown in panel C. $n = 5$ mice (mean ± SD). One-way ANOVA.

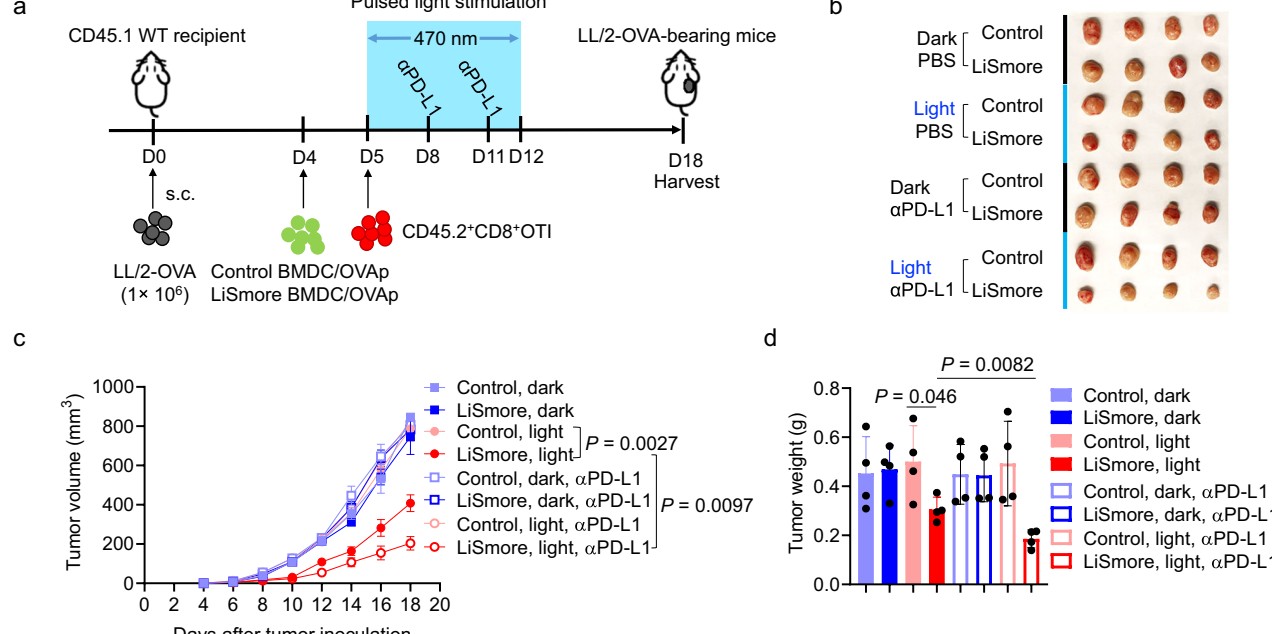

**Fig. 5 | LiSmore enhances anti-PD-L1 treatment efficacy in an immunosuppressive LL/2 lung carcinoma model. a** Schematic illustrating the in vivo LL/2 tumor model setup. $1 \times 10^6$ LL/2-OVAp lung carcinoma cells were injected (s.c.) in the flank of CD45.1 mice ($n = 4$ mice/group). Mice were transferred with OVAp-loaded BMDCs expressing Control or LiSmore at day 4, followed by adoptive transfer of CD45.2⁺ OT-1 CD8⁺ T cells. Anti−PD-L1 was administered (200 μg/mouse; i.p.) twice on days 8 and 11 after T cell transfer, using an isotype antibody as the

control. Mice were subjected to pulsed blue light stimulation (470 nm at a power density of 2 mW/cm²; 30 min ON + 30 min OFF; 6 h per day) or kept without light stimulation (dark). **b** Images of isolated tumors at day 18 for each group. **c** LL/2 tumor volumes in the indicated groups of mice ($n = 4$ per group; mean ± SD). Two-sided unpaired Student's $t$-test. **d** Quantification of tumor weights at day 18. $n = 4$ mice (mean ± SD). One-way ANOVA.

inhibited distant tumor growth while substantially reducing the occurrence of systemic side effects. Overall, these findings underscore the potential of LiSmore as a safer alternative for future personalized immunomodulation.

## Discussion

In the present study, we developed an optogenetic tool called LiSmore, which utilizes a photosensory module derived from the *Arabidopsis* photoreceptor CRY2 that exhibits superior photosensitivity to blue light. By combining CRY2 with the C-terminal tail (CTT) of STING, which acts as the scaffold to interact with the downstream TBK1 and IRF3[32], we have achieved inducible oligomerization of STING-CTT by harnessing the power of light. This, in turn, led to the recruitment of TBK1 and IRF3, as well as the formation of supramolecular organizing centers (SMOCs). Subsequent co-clustering of TBK1 results in the activation of its kinase activity to phosphorylate IRF3 and drives the nuclear translocation of IRF3, thereby inducing the expressions of type I interferons and other cytokines. To address the limited tissue

penetration of blue light, we employed a previously reported ultra-light-sensitive CRY2-clustering system called CRY2clust (A9), which incorporates a 9-residue peptide extension to the C-terminus of CRY2PHR[35]. We named this blue light-sensitive tool as LiSmore, short for "light-inducible SMOC-like repeats", to denote its ability to photo-tune STING-mediated innate immune response. LiSmore allows for rapid and efficient production of IFN-β upon biocompatible blue light stimulation, enabling wireless control of STING activity in living animals. We demonstrated the efficacy of LiSmore in vivo by showing light-triggered TBK1 and IRF3 phosphorylation in LiSmore-engineered bone marrow-derived dendritic cells (BMDCs) within the tumor-draining lymph nodes (tdLNs) of B16 melanoma-bearing mice. This successful use of blue light for remote control of STING activation in awake mice highlights the potential of LiSmore as a valuable tool for studying and manipulating STING-mediated signaling in immune cells.

Recent studies have emphasized the significance of activating the STING pathway in antigen-presenting cells to induce type I interferon production, which plays a pivotal role in the development of adaptive

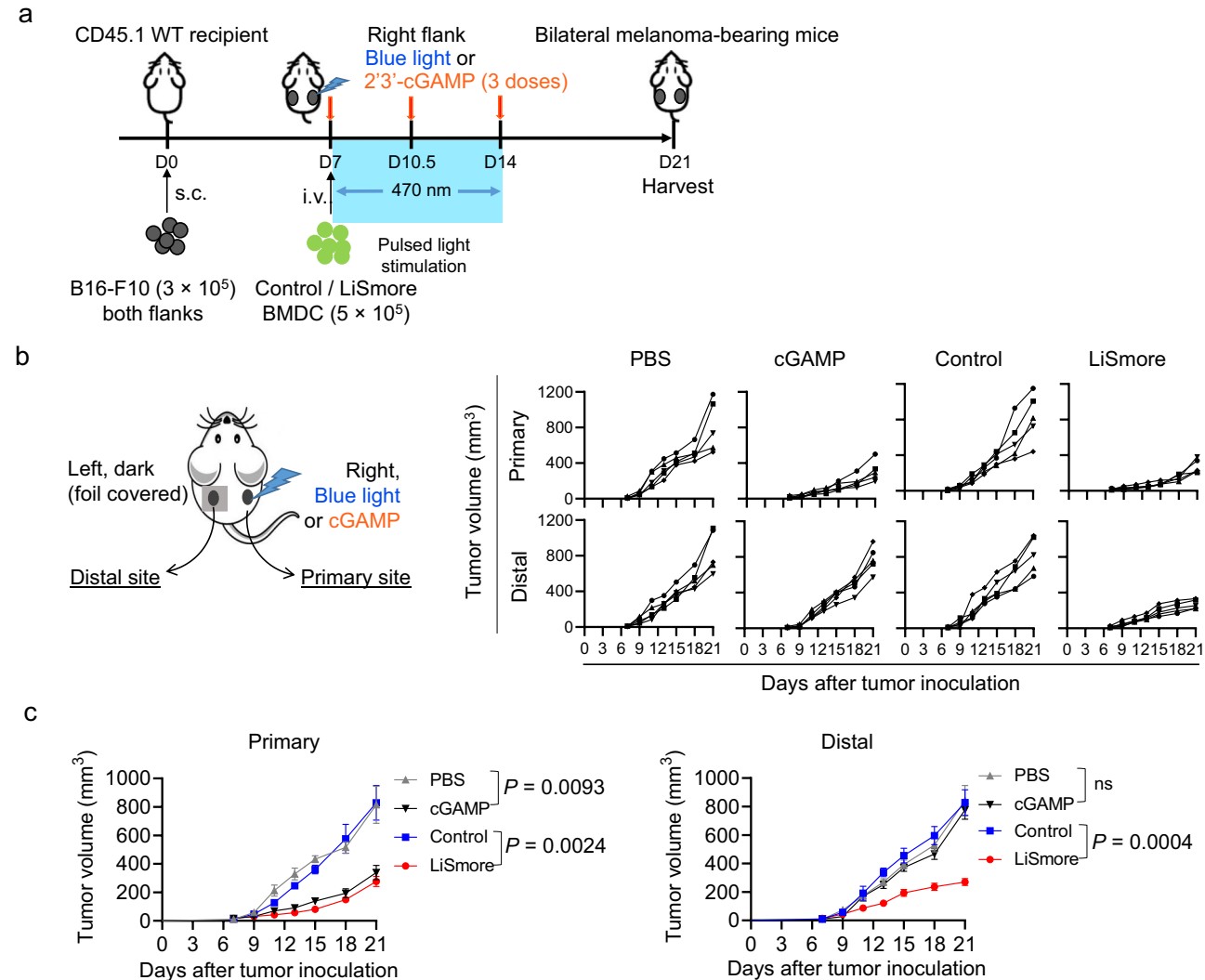

**Fig. 6 | Photo-activated LiSmore suppresses distant tumor growth in a syngeneic bilateral B16-F10 melanoma model. a** Overview of the experiment design. $3 \times 10^5$ B16-F10 melanoma cells without OVA transduction were injected (s.c.) in the right and left flanks of WT CD45.1 B6 mice. Mice received intratumoral PBS, or cGAMP (10 µg/mouse) on the right side at day 7, 10.5, and 14 after tumor inoculation, or were transferred with BMDCs expressing Control or LiSmore ($5 \times 10^5$ cells/mouse) at day 7. The right side of the tumor (the primary site) was subjected to pulsed blue light stimulation for 7 days (470 nm; 2 mW/cm$^2$; 20 s ON + 5 min OFF; 6 h per day), while the left side (the distal site) was shielded from blue light (dark). **b** Left, schematic showing the treatment in the bilateral melanoma model. Right, individual growth curves of tumors in the primary (upper panels) and distant sites (lower panels) for the indicated treatment groups ($n = 5$ mice per group). **c** Primary and distal tumor growth in the indicated groups of mice ($n = 5$ mice per group; mean ± SD). Two-sided unpaired Student's $t$-test.

immunity against malignancies[60]. In the same vein, we introduced LiSmore into dendritic cells to achieve precise control over STING-like immune response in mouse models of melanoma and lung carcinoma. We have shown that LiSmore in DCs could act as potent adjuvants, both in vitro and in vivo, by enhancing DC maturation and facilitating the cross-presentation of surrogate tumor antigens to prime effector T cells in the tumor-draining lymph nodes. As a result, this leads to antigen-specific cytotoxic CD8[+] T-cell responses and the induction of protective antigen-specific CD8[+] T-cell responses in mice, ultimately contributing to the development of robust anti-cancer immunity. Furthermore, the combination of LiSmore with anti-PD-L1 checkpoint immunotherapy yielded more promising results. Upon photostimulation, LiSmore-DCs effectively inhibited tumor growth in a PD-L1 insensitive LL/2 lung cancer model. This encouraging outcome highlights the ability of LiSmore to remodel the tumor microenvironment, effectively converting immunologically "cold" tumors into "hot" ones that are more susceptible to clearance.

Although the in-situ immunotherapy with STING agonist is appealing, clinical trials have shown limited efficacy of cGAMP

treatment, probably due to the suppression of the STING pathway within the tumor microenvironment[61]. For instance, some tumors may harbor mutations or epigenetic alterations that impair the STING pathway or render them less responsive to STING agonists like cGAMP. Moreover, restricted accessibility to specific tissues and cell populations could further hinder its effectiveness[62]. Additionally, the relatively short half-life of cGAMP poses challenges in achieving sustained activation of STING signaling[9], necessitating repeated intratumoral injections for therapeutic efficacy and limiting its potential for future clinical applications[63]. It is worth noting that excessive intratumoral cGAMP can induce T- and B-cell apoptosis[21,22], upregulate the expression of immunosuppressive molecules (such as PD-L1 and IDO1)[2,20], and promote the proliferation of tumor-infiltrating regulatory T cells, hence compromising the overall antitumor immunity[64,65]. Although the recent development of multivalent long-lasting STING agonists has partially mitigated these concerns[59,62], STING agonists can still trigger undesired immune responses, leading to the production of pro-inflammatory cytokines and the recruitment of various immune cells to the site of treatment. While this immune response can be beneficial

for combating cancer, it may also result in collateral damage to healthy tissues, leading to autoimmune disorders or other adverse effects[9]. Therefore, there is a pressing need for more precise modulation of STING signaling. In light of these challenges, our study introduces LiSmore, an optogenetic tool that enables remote activation specifically in engineered immune cells using non-invasive blue light. Our study sets the stage for future rigorous testing of LiSmore in additional pre-clinical animal models. This approach is anticipated to provide improved spatiotemporal control over engineered therapeutic immune cells, thereby minimizing off-tumor cross-reactions and mitigating undesired toxicity (Fig. 7).

Our investigations have demonstrated that LiSmore leads to more rapid and sustained activation of downstream effectors, including TBK1 and IRF3 (Fig. 2c and Supplementary Fig. 11). This prolonged activation may explain the disparities observed in CCR7, MHC-I, and T-cell proliferation in vitro (Fig. 3c), as well as the improved antigen presentation and CD80 expression within DCs in vivo (Fig. 4e, f), between the LiSmore and cGAMP treatment groups. STING activation has been shown to be crucial for the generation of stem-like central memory CD8⁺ T cells[66]. Our LiSmore tool may promote T cell memory by employing pulses of STING activation, whereas the continuous activation of STING by cGAMP may lead to T cell death due to over-activation[21,67]. Although LiSmore and cGAMP treatments resulted in comparable levels of tumor clearance in one of our tumor models (Fig. 4d), LiSmore further demonstrated abscopal anti-tumor efficacy in the bilateral tumor model (Fig. 6). Follow-on studies using tumor challenge models or metastasis models may provide deeper insights into the differential effectiveness of LiSmore and cGAMP in promoting anti-tumor immunity. These models can help evaluate the ability of the

treatments to prevent tumor growth or metastasis, as well as their long-term effects on T-cell memory and overall survival.

Through side-by-side comparisons between LiSmore-DCs and cGAMP, we have uncovered several notable advantages of the optogenetic approach (Fig. 7 and Supplementary Table 1). Firstly, LiSmore induces fewer systemic side effects than cGAMP in mouse models used in this study. Secondly, LiSmore exhibits the additional benefit of generating abscopal effects to curtail tumor growth at distal sites. This advantageous feature can be attributed, at least in part, to the prolonged activation of the STING pathway exclusively in engineered dendritic cells upon light stimulation. Lastly, LiSmore enables reversible, localized photo-activation of engineered dendritic cells, providing tight spatial control over the innate immune response. While the optogenetic approach, like other forms of adoptive cell therapies, does present challenges, such as the complex and costly procedures involved in generating engineered dendritic cells and the intricacies of light delivery, recent advancements in the field offer promising solutions to overcoming these limitations. The use of virus-like particles, exosomes, and lipid nanoparticles has shown promising progress in targeted in vivo delivery of genes of interest or the CRISPR/sgRNA complex to specific cell types[68–70]. Despite these considerations, LiSmore represents a highly promising approach for precise and reversible control of the STING pathway in cancer immunotherapy. By exclusively delivering light to tumor regions, our approach has the potential to minimize unwanted immune activation and improve the safety of immunotherapy.

In conclusion, this study introduces LiSmore as an optogenetic tool that provides precise control over STING signaling, thereby promoting dendritic cell-mediated immune sensing and cross-priming of CD8⁺ T cells. This light-switchable approach enhances tumor-specific

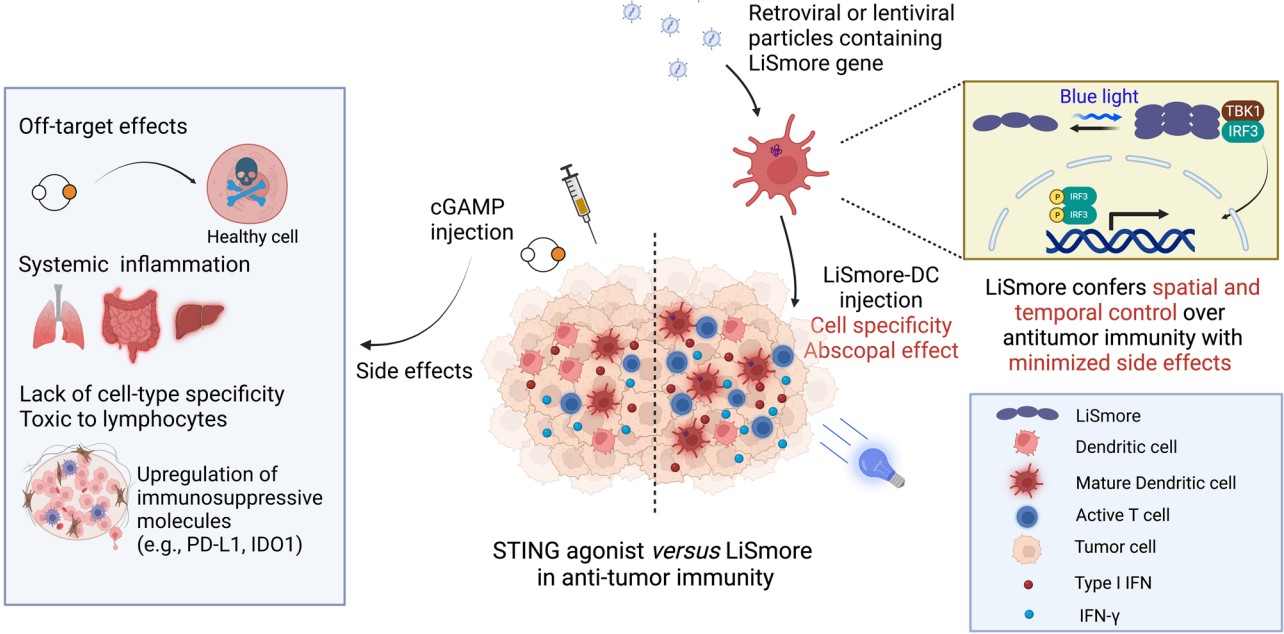

**Fig. 7 | Comparison between LiSmore and STING agonist treatment.** STING agonists, such as cGAMP, have demonstrated promising antitumor efficacy; however, their use carries the risk of systemic inflammation due to excessive cytokine production and lacks specificity in tumor targeting. Furthermore, excessive cGAMP can induce T- and B-cell death and upregulate the expression of immunosuppressive genes, such as programmed death-ligand 1 (PD-L1) and indoleamine 2,3-dioxygenase 1 (IDO1), counteracting its tumor-suppressive effects. These concerns can be addressed by developing a switchable STING activator, such as LiSmore, which utilizes light to reversibly trigger STING pathway activation, providing precise control over antitumor immunity. Local or systemic infusion of LiSmore-DCs allows for targeted STING activation through simple blue light stimulation. This precise control over STING activation leads to enhanced DC maturation and improved cross-presentation of tumor antigens, thereby priming effector T cells in tumor-draining lymph nodes to boost tumor cell killing. Additionally, LiSmore induces a systemic antitumor response through an abscopal effect, likely due to its ability to sustain STING pathway activation and elicit a more durable immune response. This unconventional strategy holds promise for improving cancer immunotherapies by harnessing the power of the STING pathway in a controlled and selective manner. Created by biorender.com.

immune response in a controllable manner. By establishing LiSmore as a photo-activatable adjuvant-like actuator, we offer promising prospects for the development of intelligent immunotherapies and smart vaccines. These optogenetic immunomodulatory tools not only facilitate the mechanistic interrogation of innate immune signaling pathways, but also create new avenues for fine-tuning immune activation while mitigating excessive inflammation[24,71,72]. This holds the potential to advance the field of immunotherapy by enabling more targeted and safer treatment strategies against cancer and immunoinflammatory disorders.

## Methods

### Cell lines
HeLa, HEK293T, THP-1, and J774A.1 cell lines were obtained from the American Type Culture Collection (ATCC, Manassas, VA, USA). HeLa, HEK293T, and J774A.1 cells were cultured in complete DMEM (Sigma-Aldrich, St. Louis, MO, USA), while THP-1 cells were cultured in complete RPMI-1640 (Sigma-Aldrich, St. Louis, MO, USA). Both media were supplemented with 10% FBS (Omega, Tarzana, CA, USA), 100 U/ml penicillin/streptomycin, 4 mM L-glutamine, and 20 mM HEPES (Invitrogen-Gibco, Carlsbad, CA).

The B16-F10 cells, as well as the B16-OVA mouse melanoma cells expressing the ovalbumin (OVA) epitope (gifts from Dr. Anjana Rao, La Jolla Institute for Immunology), were cultured in complete DMEM, and passaged at least two times prior to injection. To generate a stable LL/2 cell line expressing OVA (LL/2-OVA), a lentiviral expression vector pLVX-puro-cOVA-IRES-BFP (Addgene #135074, Watertown, MA, USA) containing the cDNA encoding full-length OVA protein was utilized. Cells expressing blue fluorescent protein (BFP) were selected through multiple rounds of fluorescence-based cell sorting. The cells were minimally passaged and maintained in complete DMEM supplemented with 10% FBS, 100 U/ml penicillin/streptomycin, 4 mM L-glutamine, 20 mM HEPES, and 10 μg/ml puromycin (Invitrogen-Gibco, Carlsbad, CA).

### Mice
All animal studies were approved by the Institutional Animal Care Use Committee (IACUC) of the Texas A&M University Institute of Biosciences and Technology (Protocol # 2021-01850-IBT). The maximum allowed tumor diameter was 20 mm.

C57BL/6-Tg (TcraTcrb) 1100Mjb/J (CD45.2, H-2b) (OT-I) mice (Strain #: 003831), C57BL/6-CD45.1 (B6 CD45.1, Strain #: 002014) and C57BL/6J (B6 CD45.2, Strain #: 000664) mice were purchased from the Jackson laboratory (Bar Harbor, ME, USA). Age- and sex-matched mice between 6 and 8 weeks of age were used in the experiments. All mice were maintained at the Institute of Biosciences and Technology, Texas A&M University (Houston, TX, USA) in specific pathogen-free/SPF conditions under standard conditions (23–26 °C, 40%–60% humidity, and 12 h light-dark cycle).

### Reagents and antibodies
The enhanced chemiluminescence (ECL) western blotting substrate was purchased from Thermo Fisher Scientific (#32106, Waltham, MA, USA). KOD Hot Start DNA polymerase (#71086-4) was purchased from Sigma-Aldrich (St. Louis, MO, USA). The T4 DNA ligase kit (#M0202M) and NEBuilder HiFi DNA Assembly Master Mix (#M5520AA) were purchased from New England BioLabs (Ipswich, MA, USA). The Quik-Change Multi Site-Directed Mutagenesis Kit (#210513) was obtained from Agilent Technologies (Santa Clara, CA, USA). Antibodies are described in Supplementary Table 2. The rabbit polyclonal anti-mCherry (NBP2-25157) antibody was obtained from Novus Biologicals (Littleton, Colorado, USA), while the mouse anti-β-Actin (sc-47778) antibody was purchased from Santa Cruz Biotechnology (Dallas, TX, USA). Secondary antibodies, including goat anti-mouse IgG−HRP (sc-2005) and goat anti-rabbit IgG-HRP (sc-2004), were purchased

from Santa Cruz Biotechnology (Dallas, TX, USA). TBK1/NAK (D1B4) Rabbit mAb #3504, Phospho-TBK1/NAK (Ser172) XP Rabbit mAb (#5483), IRF-3 (D6I4C) XP Rabbit mAb (#11904), Phospho-IRF-3 (Ser396) (4D4G) Rabbit mAb (#4947), NF-κB p65 (D14E12) XP Rabbit mAb (#8242), and Phospho-NF-κB p65 (Ser536) (93H1) Rabbit mAb (#3033) were from Cell Signaling Technology (Danvers, MA, USA). Recombinant human interferon type gamma (IFN-γ) (#300-02), recombinant human interleukin 4 (IL-4) (#200-04), and recombinant human interleukin 13 (IL-13) (#200-13) were from Peprotech. Lipopolysaccharides from *Escherichia coli* were purchased from Sigma-Aldrich (#L2630; St. Louis, MO, USA).

### Plasmids
The plasmids pDONR223-TBK1-WT (#82285), pTRIP-GFP-IRF3 (#127663), STING-V1 (#124262), pMSCV-IRES-GFP-MyD88-CpLxIS (#131348), and the packing vectors pMD2.G (#12259) and psPAX2 (#12260), as well as the lentiviral vector pWPXL (#12257, Addgene), were obtained from Addgene (Watertown, MA, USA). To generate light-inducible constructs to recapitulate STING activation, two copies of cDNA encoding the C-terminal tail of human STING (STING-CTT) and/or the mouse STING were inserted into the mCherry-CRY2PHR or mCherry-CRY2clust vectors by using the NEBuilder HiFi DNA Assembly Master Mix kit (NEB, (Ipswich, MA, USA). TBK1-YFP was made by inserting the cDNA encoding human TBK1 (#82285, Addgene) into the pEYFP-N1 vector using the KpnI and AgeI restriction sites. To generate stable cell lines, constructs containing both mCherry-CRY2PHR-pLxIS and mCherry-CRY2clust-pLxIS, along with their respective controls (mCherry-CRY2PHR and mCherry-CRY2clust), were inserted into pWPXL between the BamHI and EcoRI sites. All constructs were validated through Sanger DNA sequencing. In a further modified version, the pMSCV-GFP-IRES-CRY2clust-pLxIS plasmid, along with the empty vector pMSCV-GFP-IRES-CRY2clust-stop as control, was used for retroviral transduction of isolated mouse dendritic cells.

### Cell transfection and photostimulation
Lipofectamine 3000 reagent (Invitrogen, Waltham, MA, USA) was used for transient transfection of HeLa and HEK293T cells by following the manufacturer's instructions. For in vitro light stimulation, cells were stimulated at 24 h after transfection with a 30 s ON and 30 s OFF blue light pulse (470 nm, 0.1–4 mW/cm²; ThorLabs, Newton, NJ, USA) unless otherwise noted. Light cycles were programmed by connecting to a DC2100 LED driver with pulse modulation (ThorLabs, Newton, NJ, USA). The light intensity was measured by using an optical power meter from ThorLabs (Newton, NJ, USA). For some experiments, HEK293T cells expressing the indicated constructs were flow cytometry sorted by mCherry fluorescence first, and then seeded with matched mCherry expression (>90%) for gene expression and IFN-β detection.

### Confocal imaging
HeLa cells were plated on glass-bottomed dishes (#D35-20-0-TOP, Cellvis, Mountain View, CA, USA). Cells were then transfected with the indicated plasmids by using Lipofectamine 3000 (Invitrogen, Waltham, MA, USA). Twenty-four hours post-transfection, the samples were mounted onto a Nikon Ti2 Inverted microscope equipped with a Yokogawa W-1 dual spinning disk scan-head, Micro-Scanner for photostimulation, and a stage top incubator for live cell imaging. Blue light stimulation was carried out using the built-in 488-nm laser source with 5% input at an interval of 5 s for 5–20 min. To monitor the recruitment of TBK1-YFP, HeLa cells were transfected with plasmids encoding CRY2-pLxIs or mCh-CRY2 (200 ng) and TBK1-YFP (200 ng). 488 nm and 561 nm laser sources were used to excite TBK1-YFP and mCh-CRY2-pLxIs (or mCh-CRY2), respectively. In the experiment for monitoring the reversible clustering of LiSmore, cells were repeatedly illuminated

in two dark-light cycles (5 s ON + 10 min OFF). The captured images were analyzed by the Nikon Elements imaging processing software (Nikon, NIS-element AR version 4.0, Minato City, Tokyo, Japan) or the ImageJ program (NIH). All imaging data shown were representative of at least three biological replicates.

### In vitro BMDC induction

Murine bone marrow-derived dendritic cells (BMDCs) were obtained from 6 to 8-week-old C57BL/6 (abbreviated as B6) mice. To isolate BMDCs, bone marrow cells were flushed from the femurs of B6 mice and cultured in a 10-cm petri dish for 6 days. The culture medium consisted of DMEM supplemented with 10% heat-inactivated FBS, 100 U/ml penicillin/streptomycin, and 20 ng/ml GM-CSF (PeproTech, Cranbury, NJ, USA). The culture medium was replenished every 2 days, and the non-adherent and loosely adherent immature DCs were collected and phenotyped by determining the expression of CD11c (routinely ~90% CD11c$^+$). BMDCs were then collected on day 6 and seeded into 6-well plates for further characterization and immunophenotypic analysis.

### Retroviral transduction of BMDCs

To express LiSmore in BMDCs, retroviral plasmids (pMSCV-GFP-IRES-CRY2clust-pLxIS and pMSCV-GFP-IRES-CRY2clust-stop as the control) were first transfected into Plat-E cells (#RV-101; Cell Biolabs, San Diego, California, USA) using Lipofectamine 3000 (Invitrogen, Waltham, MA, USA). Retroviral stocks were collected twice, with the first collection starting 48 h after transfection and subsequent collection at 24-h intervals. The retrovirus-containing media were concentrated, and BMDCs were infected twice with the retrovirus in the presence of 10 μg/ml polybrene (EMD Millipore, Burlington, MA, USA). To increase the transduction efficiency in some experiments, we repeated the transduction on consecutive days[73]. Forty-eight hours after the 2nd transfection, GFP and CD11c double-positive cells were sorted using a FACSFusion cell sorter (BD Biosciences, Franklin Lakes, NJ, USA) for in vitro experiments or in vivo adoptive transfer.

### Mouse T-cell isolation for adoptive transfer

CD8$^+$ T cells were harvested from spleens and lymph nodes of OT-1 B6 mice and purified using the mouse CD8a$^+$ T-Cell Isolation Kit (Miltenyi Biotec, San Diego, CA, USA) following the manufacturer's instructions.

### Flow cytometry analysis

Single-cell suspensions from either tissues or cell culture were kept on ice and stained with Zombie Aqua Fixable viability Kit (#423102, BioLegend, San Diego, CA, USA), and blocked with anti-CD16/32 (93, Fc block, BioLegend). Surface staining was performed by incubating the cells with the appropriate antibodies for 20 min on ice. Antibodies are described in Supplementary Table 3. The following antibodies from BioLegend were used: PerCP/Cy5.5-CD45 (30-F11), Alexa Fluor 700-CD45.2 (104), Percp-CD11c (N418), PE-H2Kb (28–14–8), APC-IA/IE (M5/114.15.2), APC-CD86 (GL-1), PE-CD80 (16-10A1), APC-CD40 (3/23) and PE-CCR7 (4B12). OT-1 CD45.2$^+$CD8$^+$ TILs were stained with Alexa Fluor 700-CD45.2 (104), PE/Cy7-CD45.1 (A20), APC-CD8α (53.6-7), PE-CD69 (H1.2F3), APC anti-human CD80 (2D10), and Brilliant Violet 421 anti-human CD86 (IT2.2). For intracellular staining of IFN-γ, OT-I CD8 T cells were incubated for 4 h at 37 °C in the presence of monensin (BD Biosciences, Franklin Lakes, NJ, USA). After surface staining, cells were permeabilized using cytofix/cytoperm (BD Biosciences) for 30 min on ice. Permeabilized cells were then resuspended in BD Perm/Wash buffer (BD Biosciences) and stained with a PE-anti-IFN-γ (XMG1.2; BioLegend) antibody for 30 min. PE-anti-Ki67 (16A8) staining was done using a Foxp3 / Transcription Factor Staining Buffer Set (00–5523, eBioscience, San Diego, CA, USA) according to the manufacturer's

instructions. STING activation was assessed by staining Alexa Fluor 555-p-TBK1 (D52C2, CST, Danvers, MA, USA) and Alexa Fluor 647-p-IRF3 (D6O1M, CST, Danvers, MA, USA) in CD45.2$^+$GFP$^+$ DCs with BD Phosflow buffer (#557870, BD Biosciences, Franklin Lakes, NJ, USA). Cells were also stained with matched isotype control antibodies, including Alexa Fluor 555-Rabbit mAb IgG (DA1E, #3969, CST) and Alexa Fluor 647-Rabbit mAb IgG (DA1E, #2985, CST). All flow cytometry data were collected using a LSRII flow cytometer (BD Biosciences, Franklin Lakes, NJ, USA) and analyzed using the FlowJo software (Ashland, OR, USA).

### RT-PCR and quantitative PCR analysis

Total RNA was isolated using TRIzol reagent (Invitrogen, Waltham, MA, USA) according to the manufacturer's instructions. cDNA was synthesized from total RNA using a Superscript III First-Strand cDNA synthesis kit (Invitrogen, Waltham, MA, USA). Real-time PCR was performed using an ABI PRISM cycler (Life Technologies, Carlsbad, CA, USA) with a SYBR Green PCR kit from Applied Biosystems (Life Technologies, Carlsbad, CA, USA). The resulting data were presented as the accumulation index ($2^{\Delta\Delta Ct}$). The primer pairs used in the assay were obtained from Integrated DNA Technologies (Coralville, IA, USA) and are listed in Supplementary Table 4.

### Western blot analysis

Cells were washed three times with chilled PBS and lysed directly using Pierce IP lysis buffer (#87788, Thermo Fisher Scientific, Waltham, MA, USA) for 30 min at 4 °C. The lysis buffer contained 1x protease inhibitor cocktail (#P3100-010, GenDEPOT, Katy, TX, USA) and phosphatase inhibitor cocktail (#P3200-001, GenDEPOT, Katy, TX, USA). After lysis, the samples were denatured at 95 °C for 10 min and loaded onto an 8-16% gradient SDS-PAGE (GenScript, Piscataway, NJ, USA) along with 1x SDS loading buffer (100 mM Tris-HCl, 4% SDS, 0.2% bromophenol blue, 20% glycerol, 200 mM DTT, pH 7.4). Proteins were transferred onto nitrocellulose membranes (Bio-Rad, Hercules, CA, USA) and blocked in 5% BSA for 1 h at room temperature. The membranes were then incubated with the corresponding primary antibodies overnight at 4 °C, followed by incubation with secondary antibodies at room temperature for 1 h. The antigen-antibody complexes were visualized using the ChemiDoc Imaging System (Bio-Rad) with West-Q Pico Dura ECL Solution (GenDEPOT, Katy, TX, USA).

### Cytokine detection

For IFN-α/β detection, GFP$^+$CD11c$^+$ transduced BMDCs at a density of $1 \times 10^6$ cells/ml were seeded into 48-well plates. The supernatants were harvested after 16–18 h of incubation under pulsed blue light (470 nm, 20 s ON, 5 min OFF, 1–4 mW/cm$^2$) or in the dark. IFN-α and IFNβ were quantified using the IFN-alpha/ IFN-beta bioluminescent ELISA kit (luex-mifnav2, luex-mifnbv2; InvivoGene, San Diego, CA, USA) following the manufacturer's instructions. The absorbance was measured using a Synergy Neo2 microplate reader (BioTek, Winooski, VT, USA). Additionally, inflammatory cytokines in supernatants were analyzed by the mouse LEGENDplex custom flow analyte kit (BioLegend) and subjected to flow cytometry analysis using a LSRII flow cytometer. The concentrations of IFN-α and IFN-γ in tumor homogenates and serum were measured using the IFN-α and IFN-γ ELISA Kit (KMC4021, Invitrogen).

### Peptide pulsing of transduced BMDCs

For peptide pulsing, purified GFP$^+$ BMDCs at a density of $1 \times 10^6$/ml were resuspended in DMEM containing 10 μg/ml OVA (257–264) peptide (Sigma-Aldrich, St. Louis, MO, USA). After 3 h incubation at 37 °C with gentle shaking every 30 min, the OVAp-pulsed BMDCs were washed twice with PBS and resuspended in PBS for vaccination of the mice.

## In vitro BMDCs maturation and cross-presentation assay

In vitro differentiated BMDCs were collected on day 6 and seeded into 6-well plates for transduction. On day 7 and day 8, BMDCs were transduced with viruses encoding LiSmore or the control vector, or left untransduced. LiSmore-transduced BMDCs were harvested on day 10 for in vitro experiments. To assess BMDC maturation, the cells were subjected to flow cytometry analysis after staining with the following antibodies: Percp5.5-CD11c (N418), PE-H2Kb (28-14-8), APC-IA/IE (M5/114.15.2), APC-CD86 (GL-1), PE-CD80 (16-10A1), APC-CD40 (3/23), and PE-CCR7 (4B12). For the cross-presentation assay, B16-OVA cells were treated with mitomycin C (50 μg/ml) and co-cultured with BMDCs at a 1:1 ratio, either with or without pulsed blue light stimulation overnight (470 nm, 20 s ON, 5 min OFF, 1 mW/cm²). In selected groups, non-transduced BMDCs were treated with or without 2′3′-cGAMP (2 μg/ml) under the same timeframe and conditions. After 24 h, OVAp presented with MHC-I on the cell surface was detected using APC-conjugated anti-mouse H-2Kb bound to the SIINFEKL antibody (25-D1.16, BioLegend).

## In vitro CD8 T-cell proliferation and priming assays

Prior to co-culture with OT-I CD8$^+$ T cells, BMDCs were pulsed with 10 μg/ml OVAp for 2 h and treated with 50 μg/ml mitomycin C for 30 min at 37 °C. CD8$^+$ T cells were purified from the spleens of WT OT-I transgenic mice using a mouse CD8$^+$ T Cell Isolation Kit (Miltenyi Biotec, San Diego, CA, USA). OT-I CD8$^+$ T cells were labeled by using the CellTrace Violet Cell Proliferation Kit (BioLegend, San Diego, CA, USA). $1 \times 10^5$ labeled OT-I CD8 T cells were then added to 96-well U-bottom plates containing mitomycin C-treated and OVAp-pulsed LiSmore-BMDCs. The cells were exposed to the pulsed blue light for the first 18 h (470 nm, 20 s ON, 5 min OFF, 1 mW/cm²). In selected groups, OT-1 CD8$^+$ T cells were co-cultured with untransduced BMDCs/OVAp and treated with either 2′3′-cGAMP (2 μg/ml) or PBS as a control. The proliferation index of OT-1 CD8$^+$ T cells was analyzed on day 4 by FACS.

To evaluate antigen presentation by LiSmore-BMDCs, IFN-γ secretion by primed OT-I CD8$^+$ T cells was utilized as a measurement of CD8$^+$ T-cell activation. OT-1 CD8$^+$ T cells and LiSmore-BMDCs were prepared as described above. Briefly, OT-1 CD8$^+$ T cells were plated at a density of $2 \times 10^5$ cells/well in 96-well plates, and $2 \times 10^5$ OVAp-pulsed BMDCs were added for 18 h with or without blue light stimulation (470 nm, 20 s ON, 5 min OFF, 1 mW/cm²). Additionally, untransduced BMDCs/OVAp were co-cultured with OT-I CD8$^+$ T cells in the presence of 2′3′-cGAMP (2 μg/ml) or PBS as a control for 18 h. Cells were then collected and analyzed for Intracellular expression of IFN-γ in OT-I CD8$^+$ T cells using flow cytometry.

## Lactate dehydrogenase (LDH) release assay to assess cytotoxicity

OT-1 CD8$^+$ T cells were mixed with BMDCs and incubated with pre-plated B16-OVA cells (CD8$^+$ T: BMDC: B16-OVA = 2:1:1) in 96-well plates with either 2 μg/ml 2′3′-cGAMP (untransduced BMDCs) or blue light exposure (transduced with LiSmore, 470 nm, 20 s ON, 5 min OFF, 1 mW/cm²) for 24 h. LDH release was determined in supernatants from wells containing B16-OVA cells only, and the maximal release of LDH was determined in supernatants from wells containing the lysis solution. Supernatants from all test and control wells were collected and transferred to a fresh 96-well flat-bottom plate for the LDH release assay. The absorbance at 490 nm was recorded by using a Synergy Neo2 plate reader (BioTek, Winooski, VT, USA). Cytotoxicity (%) was calculated as (Experimental value – Background value)/(Maximal value – Background value) × 100.

## Flow cytometric quantification of cytotoxicity

For FACS-based in vitro cytotoxic assays, B16-OVA cells were first labeled with the CellTrace Violet dye (BioLegend, San Diego, CA, USA) and then cultured in triplicates with effector OT-1 CD8$^+$ T cells and

BMDCs (CD8$^+$ T: BMDC: Tumor = 2:1:1) in 96-well plates. The cells were treated as described above for the LDH release assay. After 24 h, cells were detached using trypsin and washed with cold PBS three times. Subsequently, the cells were stained with 7-AAD (#420404, BioLegend) according to the manufacturer's instructions. Dye toxicity and spontaneous B16 melanoma cell death were controlled by including a control without effector cells, and the observed levels did not exceed 5%. Stained cells were analyzed using a LSRII flow cytometer (BD Biosciences), and the data were processed using the FlowJo software. Cytotoxicity was determined by FACS analysis as the percentage of dead B16 cells (labeled as Violet$^+$7-AAD$^+$).

## Tumor models, adoptive cell transfer, and 2′3′-cGAMP / anti−PD-L1 treatment

For subcutaneous implantation of tumor cells, either B16-OVAp cells ($3 \times 10^5$ cells/mouse) or LL/2-OVAp cells ($1 \times 10^6$ cells/mouse) were trypsinized and resuspended in 100 μl PBS. The cell suspension was then injected subcutaneously (s.c.) into the flank of 6- to 8-week-old CD45.1 B6 recipient mice (both male and female, Day 0). Four days after tumor inoculation, in vitro sorted GFP$^+$ LiSmore-expressing BMDCs or GFP$^+$ Control-expressing BMDCs were loaded with OVAp (residues 257-264) and injected intravenously (i.v.) into mice ($5 \times 10^5$ cells/mouse). At day 5, CD45.2$^+$CD8$^+$ OT-I cells were retro-orbitally injected into tumor-bearing CD45.1 mice ($2 \times 10^6$ cells/mouse). For the B16 melanoma model, tumor-bearing mice were randomized into 2 ′3′-cGAMP treatment groups in addition to the BMDCs transfer groups. After OT-I CD8$^+$ cells transfer, 2′3′-cGAMP (10 μg/mouse) in 50 μl PBS was injected into the muscle (i.m.) of the hind leg on the tumor side every 3 days, starting at day 5. An equivalent amount of PBS was injected as a control.

For immune checkpoint blockade therapy in the LL/2-OVAp tumor model, 200 μg anti−PD-L1 antibody (clone 10 F.9G2, BioX-Cell, Lebanon, NH, USA) or a control anti-rat IgG2b (clone LTF-2, BioXCell, Lebanon, NH, USA) in 200 μl PBS were injected intraperitoneally (i.p.) into the corresponding groups on days 8 and 11 post-tumor inoculation. Recipient CD45.1 mice with transferred BMDCs were exposed to pulsed blue light (470 nm, 6 h per day, 30 min ON/OFF cycle, 2 mW/cm²) or shielded from blue light (control) for 7 days to stimulate LiSmore-DC activation. Tumor growth was measured on the indicated dates using a caliper, and the tumor size was calculated in mm³ using the formula: length × width² × 0.52. The maximum allowable tumor size is 20 mm in diameter for each mouse.

## A bilateral B16-F10 melanoma model to evaluate spatial control and abscopal effect

For the establishment of a syngeneic mouse model of melanoma-bearing tumors on both flanks, $3 \times 10^5$ B16-F10 melanoma cells were injected (s.c.) in the right and left flanks of 6- to 8-week-old (both male and female) CD45.1 mice. After 7 days of inoculation, the mice were divided into four groups. The mice in these groups received different treatments: intratumoral administration of PBS, or cGAMP (10 μg/mouse) on the right side of the tumor at days 7, 10, and 14 after tumor inoculation, or transfer of BMDCs expressing Control or LiSmore ($5 \times 10^5$ cells/mouse) at day 7. The mice receiving BMDC transfers were subjected to pulsed blue light stimulation for 7 days (470 nm; ~2 mW/cm²; 20 s ON + 5 min OFF; 6 h per day) on the right side of tumor (defined as the primary site). Tumors on the left flank (the distal site) were shielded from blue light. Tumor volumes of the mice were recorded continuously throughout the experiments (length × width² × 0.52). Tumor size must not exceed 20 mm in any direction in each mouse. One day after the final treatment (day 15), blood samples were collected from each mouse without heparinization, and then centrifuged for 5 min to separate sera for subsequent safety studies.

## Evaluation of systemic toxicity and inflammation

The activities of serum alanine aminotransferase (ALT) and aspartate aminotransferase (AST), as well as the serum urea level, were measured using commercially available colorimetric kits from Abcam (ab105134, ab105135, and ab83362, respectively). The mouse serum interleukin-6 (IL-6) level was determined using an enzyme-linked immunosorbent assay (ELISA) kit from ThermoFisher Scientific (KMC0061).

## DC isolation from tumor-draining lymph nodes (tdLNs)

To examine STING activation in the transferred DCs, tumor-draining lymph nodes (tdLNs) were harvested at day 5 (18 h post-photo-stimulation) and dissociated into a single-cell suspension using a 40-μm cell strainer. The cell suspension was then incubated with 1 mg/ml collagenase D and 50 μg/ml DNase I (Sigma-Aldrich, St. Louis, MO, USA) at 37 °C on a shaker for 40 min. Subsequently, the cells were washed twice in a complete medium and prepared for FACS analysis. STING activation was assessed by staining CD45.2+CD11c+ GFP+ DCs with Alexa Fluor 555-p-TBK1 (D52C2), and Alexa Fluor 647-p-IRF3 (D6O1M, Cell Signaling Technology, Danvers, MA, USA).

## Tumor-infiltrating lymphocyte (TIL) isolation

To analyze the activity of TILs, tumor-bearing mice were euthanized by carbon dioxide at day 12, which was 7 days after adoptive T cell transfer. Tumors were dissected and minced, and then digested with 1 mg/ml collagenase D and 50 μg/ml DNase I for 45 min at 37 °C with gentle shaking. The resulting cell suspension was passed through a 100-μm filter and washed with FACS buffer (PBS containing 2% FBS and 2 mM EDTA). Red blood cells were removed using ACK lysis buffer (BioLegend). The cells were resuspended in FACS buffer on ice for subsequent FACS analysis. CD69, Ki67, and IFN-γ expression in CD45.2$^+$CD8$^+$ TILs were assessed by flow cytometry.

## Data analyses

All statistical analyses were performed on Prism 7 (GraphPad Software, San Diego, CA). Statistical analysis was performed using either one-way ANOVA or two-tailed unpaired $t$-test. Quantitative data were presented as mean ± SD.

## Reporting summary

Further information on research design is available in the Nature Portfolio Reporting Summary linked to this article.

# Data availability

The data supporting the findings of this study are available within the paper and its supplementary information files. The plasmids and all other data are available from the corresponding authors upon reasonable request. Source data are provided in this paper.

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

## Acknowledgements

This work was supported by the Welch Foundation (BE-1913-20220331 to Y.Z.), the Leukemia & Lymphoma Society (LLS6680-24 to Y.Z.), and the National Institutes of Health (R01GM144986 to Y.Z., R35HL166557 to Y.H., R01CA240258 to Y.H., and R21AI174606 to Y.Z.). Mouse cartoon in Figs. 2a, 3a, 4b, 5a, and 6a are licensed under the Creative Commons Attribution-Share Alike 3.0 Unported. Mouse cartoon in Fig. 6b was from the online sharing service Clker.com. Figures 2a and 3a were drawn by using pictures from Servier Medical Art. Reproduced with permission,

licensed under a Creative Commons Attribution 3.0 unported license. Figures 1a, b, and 7 were created by Biorender.com.

## Author contributions
Conceptualization: Y.Z., Y.H., Y.D.; Experimentation: Y.D., R.C., Y.L., J.J., S.L., X.L., Y.K., R.W.; Data analysis: Y.D., R.C.; Supervision: Y.Z., Y.H.; Manuscript preparation: Y.D., R.C., Y.H., Y.Z.

## Competing interests
The authors declare no competing interests.
