## [Peer Review File · Nature Communications]

REVIEWER COMMENTS

Reviewer #1 (expert in cGAS/STING, STING signalling):

Dou et al described an optogenetic design where they can control STING activation with blue light. The design is a fusion protein of STING C-terminal IFN signaling domain (CTT) with a light-inducible oligomerizing domain (CRY2). The authors first tested this design and a derivative they call LiSmore in vitro. Both constructs can be activated by blue light, and the immune signaling is consistent with STING activation (e.g. elevated pTBK1, pIRF3, production of IFN β , etc). They further showed that the LiSmore design also work in mice by triggering similar IFN signaling when exposed to blue light. They also expressed LiSmore in BMDCs and showed that blue light activation can activate BMDCs and induce CD8-mediated tumor cell killing. In an in vivo syngeneic tumor model, LiSmore-transduced BMDC can also suppress tumor growth when exposed to blue light similar to STING agonist cGAMP, and synergy with PD-L1 treatment.

All the experiments are well executed and well controlled. Data are consistent throughout. The optogenetic design convincingly activates STING-mediated IFN signaling in vitro and in vivo similar to STING agonist cGAMP.

There is a concern on limited novelty. The fusion protein is a novel design. Everything else is rather predictable and nothing new on STING biology. Overexpression of STING CTT domain or fusion with other dimerizers are known to activate IFN signaling. Thus, using blue light to induce oligomerization through the CRY2 domain fusion not surprisingly also activate signaling.

From a practical standpoint, it is unclear this optogenetic design is better than existing STING agonists in anti-tumor therapy. In this study, the authors had to transduce BMDCs with the construct, transfer OT1 CD8 T cells, and have matching B16-OVA to show anti-tumor activity, whereas, a simple intra-tumor or systemic injection of STING agonist in native tumor-bearing mice would show a similar effect. To increase the impact of this study, the authors should demonstrate more superior utility of this design over STING agonists. E.g. systemically deliver the construct, then locally activate signaling using focused beam of blue light? Can they control activation more precisely at tumors or at inner organ mets where STING agonist is lacking? What about systemic toxicity, abscopal effects?

Reviewer #2 (expert in cancer therapy, PD1 engineering):

The manuscript entitled “Optogenetic engineering of STING signaling to remotely control anti-tumor immunity” submitted by Yaling Dou et al., describes the novel noninvasive approach for the development of cancer therapeutics. Investigators manipulate the cGAS-STING signaling pathway by using ultra-light-sensitive optogenetic LiSmore for light-inducible by remotely control STING activation and downstream gene expression in antigen-presenting cells and to activate innate immunity. This well-designed study was meticulously carried out to tackle every step of the process, primer design, cellular studies, transformation, in vitro, ex vivo and in vivo mouse studies.

Investigators systematically start this study from STING signal pathway in the cell (in vitro) and test the hypothesis in pre-clinical models with the transgenic tumor xenografted mice. This is a good example for translational research. The study design, manuscript presentation, data analysis, results and discussion and conclusion were well done. Except minor typos and edits. The depth and breadth of “noninvasive activation of LiSmore” story is quite intriguing, and I feel this manuscript could be well deserved to be published in Nature communications.

Overall, the study results, data validity and significance of the research are good quality based on this assessment I would like to recommend that this manuscript can be accepted for the publication after addressing the following comments and feedback.

Comments:

1. Although this manuscript provided all details regarding data (including SI and video) with smooth flow from figures 1-5 in a sequential manner, but it lacks eagerness to read due to overwhelming acronyms. Authors could have presented a concept figure in such a way to capture the quick overview of the entire study that may be helpful to readers, since abstract do not reflect the entire volume of work performed by the authors.
2. The primers list can be moved to SI.
3. Source of some of the mouse strain was not provided e.g., C57BL/6.
4. Figure 1E, x-axis list of acronyms needs to be expanded.
5. Light-inducible SMOC-like repeats, CRY2clust-pLxIs, was abbreviated as LiSmore, this could be printed in abstract and one more time at the beginning of the text, but this abbreviation was repeated almost in all instances both in text and figure in parenthesis why?
6. Figure 3F, in legend it was indicated Left, schematic showing the in vitro coculture system, this was missing only two sections were presented, this need to be corrected.

7. Figure 4, presentation of data was well laid. Figure 4-D, it looks displaying tumor volume shrinkage was almost equal to cGAMP and LiSmore treated, this was not explained well in main text.
8. Figure 5C-D, legend marks were common for both sections, it should be laid in center or provide separate for 5D.
9. SI, Figures S2, and S4, axis or legend labels fonts are printed not clear it is cramped, please fix it.

Minor typos:

Author contributions: It is printed as “date analysis”, it should be fixed.

Reviewer #3 (expert in optogenetics):

The authors developed an optogenetic strategy to activate STING signaling based on inducible clustering of CRY2. They also demonstrated the usage of an ultra sensitive version of CRY2 to reduce light dosage during in vivo treatments. Then they further verified the activation of downstream genes in the STING pathway after light induction, including antigen presenting capability and cytokine producing efficiency. Using the LiSmore strategy, they obtained similar tumor toxicity comparable to cGAMP treatment both ex vivo and in vivo. Finally, combining with the checkpoint immunotherapy, they improved the therapy for tumors with immune-suppressive microenvironment.

The tool the authors developed has been proposed as an alternative for cGAMP treatment with better spatio-temporal and cell-type specificity, which is promising in future therapies. The development of the protein tools and its successful application in vivo is exciting. Congratulations to the authors on impressive work. The data are convincing. We only suggest a few minor changes to improve the manuscript.

1. The manuscript would be strengthened by more clearly establishing the existing gap that this paper solves, perhaps even in the introduction. The deficiency of cGAMP treatment is only mentioned in the discussion, but would help frame the motivation of the paper.
2. Along the same lines, the authors should make a good case for why optogenetics is important for this use case. Given that perturbations are not done in a fast timescale, and that light control would be limited only to those situations where light could be administered topically, it seems that a chemically-induced protein aggregation approach might be more effective (drug is applied systemically but only

acts on the engineered DCs). What benefit does optogenetics offer here relative to other control modes?

3. Given phototoxic effects of blue light, the authors should include a negative control in 1E, for example cells transfected with Cry2-Ctrl (as in 1F)

4. The comparisons between Cry2 variants in Fig1G are only interpretable when the expression levels of the tools are the same. Were these cell lines expression matched?

5. Since 3I is a coculture of 3 cell types, can the authors use a more specific assay to verify death of the B16-OVA? It seems LDH release only informs that some cells are dying, not which ones.

6. Since the authors use cGAMP stimulation as a positive control throughout the paper, it would be helpful to see the effect of cGAMP on pTBK and pIRF3, for example in figure 2. Could differences in pTBK and pIRF3 between LiSmore and cGAMP explain later differences in CCR7 level, MHC-1 level and T-cell proliferation?

7. The authors should discuss why LiSmore and cGAMP might give equivalent levels of tumor clearance despite better antigen presentation capacity and higher CD80 of the LiSmore DCs compared to cGAMP. Are there other pathways controlled by STING that might drive the differences?

8. For Experiments in Figure 2 and Figure 3, can the authors comment on why they transduce cells on consecutive days? Is this necessary?

9. The text in the supplementary figures often rendered incorrectly

Point-to-point responses

We would like to express our gratitude to the editor and all three reviewers for your insightful and positive comments, which have significantly improved our manuscript. We have considered all the points raised and have conducted all the recommended experiments. We have also made clarifications to address the questions raised by all three Reviewers. All changes made in the main text and supplementary materials were shown in **blue font**. A table summarizing the new and revised figures were appended below for your reference (**Table A1**).

Table A1. Major new data added in the revised manuscript.

Figure / panel No.	Original Figure No.	Response to Reviewer(s)	Remarks
1E	1E	R3.3	Added the CRY2-control group
2C	2C	R3.6	Side-by-side comparison: Control, LiSmore and cGAMP WB: p-TBK and p-IRF3 time-course
6	New	R1, R2, R3	LiSmore and cGAMP comparison: abscopal effects
7	New	R1, R2.1	Conceptual figure as advised
S3	New	R1, R3.2	Demonstration of reversibility
S6	New	R3.5	Specific cell death in the target cells by an independent FACS-based assay
S10	New	R1	Spatial controllability as advised
S11	New	R1	LiSmore and cGAMP comparison: systemic anti-tumor responses
S12	New	R1	In vivo biosafety evaluation of LiSmore versus cGAMP as advised
Table S1	New	R2.2	Primers summarized in a new table
Table S2	New	R1, R2, R3	Comparison between LiSmore and cGAMP treatments.

Reviewer 1 (expert in cGAS/STING, STING signalling):

Dou et al described an optogenetic design where they can control STING activation with blue light. The design is a fusion protein of STING C-terminal IFN signaling domain (CTT) with a light-inducible oligomerizing domain (CRY2). The authors first tested this design and a derivative they call LiSmore in vitro. Both constructs can be activated by blue light, and the immune signaling is consistent with STING activation (e.g. elevated pTBK1, pIRF3, production of IFN β , etc). They further showed that the LiSmore design also work in mice by triggering similar IFN signaling when exposed to blue light. They also expressed LiSmore in BMDCs and showed that blue light activation can activate BMDCs and induce CD8-mediated tumor cell killing. In an in vivo syngeneic tumor model, LiSmore-transduced BMDC can also suppress tumor growth when exposed to blue light similar to STING agonist cGAMP, and synergy with PD-L1 treatment.

All the experiments are well executed and well controlled. Data are consistent throughout. The optogenetic design convincingly activates STING-mediated IFN signaling in vitro and in vivo similar to STING agonist cGAMP.

There is a concern on limited novelty. The fusion protein is a novel design. Everything else is rather predictable and nothing new on STING biology. Overexpression of STING CTT domain or fusion with other dimerizers are known to activate IFN signaling. Thus, using blue light to induce oligomerization through the CRY2 domain fusion not surprisingly also activate signaling.

Responses to R1 comments

We sincerely thank the reviewer for the supportive remark that “*All the experiments are well executed and well controlled. Data are consistent throughout.*”

1.1. “*It is unclear this optogenetic design is better than existing STING agonists in anti-tumor therapy.*”

Response: We deeply appreciate the reviewer for this insightful suggestion to investigate the potential advantages of our optogenetic design over existing STING agonists in anti-tumor

therapy (as summarized in **Table S1** and **Figure 7**). Although exhibiting encouraging anti-tumor activity, systematic use of STING agonists has been reported to have the following issues:

- i. cGAMP Treatment has shown limited efficacy in clinical trials, likely due to the suppression of STING pathway in tumor microenvironment (PMID: 33979578). For instance, some tumors may possess mutations or epigenetic alterations that impair the STING pathway or render them less responsive to STING agonists like cGAMP. Moreover, restricted accessibility to specific tissues and cell populations could further hinder its effectiveness (PMID: 36442099).
- ii. The relatively short half-life of cGAMP presents challenges in achieving sustained activation of STING signaling (PMID: 35107989), necessitating repeated intratumoral injections for therapeutic efficacy, while may not be feasible or practical for many clinical cases (PMID: 30545676).
- iii. cGAMP treatment lacks cell-type and tissue specificity. Since STING is broadly expressed in normal tissues and also tumors, it may trigger off-target tissues activation and systemic toxicities. Recent studies have shown STING overstimulation can be toxic to lymphocytes (causing T cell and B cell death) (PMID: 30886058, PMID: 26951929).
- iv. STING agonists have the potential to induce the expression of immunosuppressive molecules, such as programmed death-ligand 1 (PDL1) and indoleamine 2, 3-dioxygenase 1 (IDO1), which could readily counteract the tumor-suppressive effects (PMID: 33833439, PMID: 30170810).
- v. Although the recent development of multivalent long-lasting STING agonists has partially mitigated the above concerns (PMID: 33558734, PMID: 36442099), STING agonists can still trigger undesired immune responses, leading to the production of pro-inflammatory cytokines and the recruitment of various immune cells to the site of treatment. While this immune response can be beneficial for combating cancer, it may also result in collateral damage to healthy tissues, leading to autoimmune disorders or other adverse effects (PMID: 35107989).
- vi. In light of these challenges, our study introduces LiSmore, an optogenetic tool that enables remote activation specifically in engineered immune cells using non-invasive blue light. Our study paves the way for future testing of LiSmore in additional pre-clinical animal models and its translational application in cancer immunotherapy. This approach is anticipated to provide improved spatiotemporal control over engineered therapeutic immune cells, thereby minimizing off-tumor cross-reactions and mitigating undesired toxicity (**Figure R1**, new **Figure 7**).

Advantages of LiSmore over cGAMP. In most ex vivo experiments, the LiSmore strategy outperformed cGAMP treatment. LiSmore led to more rapid and sustained activation of downstream effectors, including TBK1 and IRF3 (**Figure 2C**, and **Figure S11**). This prolonged activation may explain why LiSmore promoted more expression of CCR7 and MHC-I in dendritic cells to enhance T-cell proliferation (**Figure 3 C**), as well as the improved antigen presentation and CD80 expression within DCs in vivo (**Figure 4E-F**). More importantly, we conducted a side-by-side comparison of the antitumor efficacy in three different mouse models, the B16-OVA melanoma model, the LL/2 lung carcinoma model, and the newly-added bilateral melanoma model (without OVA transduction and more wild-type like model). LiSmore and cGAMP treatments resulted in comparable levels of tumor clearance in the B16-OVA mouse model (**Figure 4D**). However, LiSmore demonstrated abscopal anti-tumor efficacy in the bilateral tumor model of melanoma (**Figure 6**), which was not observed for the cGAMP group. Through side-by-side comparisons between LiSmore-DCs and cGAMP, we have uncovered several notable advantages of the optogenetic approach (**Figure 7**, and **Table S2**). Firstly, LiSmore induces fewer systemic side effects than cGAMP in mouse models used in this study. Secondly, LiSmore exhibits an additional benefit of generating abscopal effects to curtail tumor growth at distal sites. This advantageous feature can be attributed, at least in part, to the prolonged activation of the STING pathway exclusively in engineered dendritic cells upon light stimulation. Lastly, LiSmore enables reversible, localized photo-activation of engineered dendritic cells, providing tight spatial control over innate immune response.

However, the optogenetic approach, like other forms of adoptive cell therapies, does presents challenges, such as the complex and costly procedures involved in generating engineered dendritic cells and the intricacies of light delivery. Recent advancements in the field could offer promising solutions to overcoming these limitations. The use of virus-like particles, exosomes, and lipid nanoparticles has shown promising progress in targeted in vivo delivery of genes of interest or the CRISPR/sgRNA complex to specific cell types (PMID: 36516854, PMID: 36516854, PMID: 34990237).

Despite these considerations, LiSmore represents a highly promising approach for precise and reversible control of the STING pathway in cancer immunotherapy. By exclusively delivering light to tumor regions, our approach has the potential to minimize unwanted immune activation and improve the safety of immunotherapy.

A brief summary of these points is presented in the **Figure R1** (or new **Figure 7**) below, as well as in **Table R1** (or new **Table S2**).

Figure R1. Comparison between LiSmore and STING agonist treatment.

STING agonists, such as cGAMP, have demonstrated promising antitumor efficacy; however, their use carries the risk of systemic inflammation due to excessive cytokine production and lacks specificity in tumor targeting. Furthermore, excessive cGAMP can induce T and B cell death and upregulate the expression immunosuppressive genes, such as programmed death-ligand 1 (PD-L1) and indoleamine 2, 3-dioxygenase 1 (IDO1), counteracting its tumor-suppressive effects. These concerns can be addressed by developing a switchable STING activator, such as LiSmore, which utilizes light to reversibly trigger STING pathway activation, providing precise control over antitumor immunity. Local or systemic infusion of LiSmore-DCs allows for targeted STING activation through simple blue light stimulation. This precise control over STING activation leads to enhanced DC maturation and improved cross-presentation of tumor antigens, thereby priming effector T cells in tumor draining lymph nodes to boost tumor cell killing. Additionally, LiSmore induces a systemic antitumor response through an abscopal effect, likely due to its ability to sustain STING pathway activation and elicit more durable immune response. This unconventional strategy holds promise for improving cancer immunotherapies by harnessing the power of the STING pathway in a controlled and selective manner. Created by biorender.com.

Table R1. Comparison between LiSmore and cGAMP.

Parameters	cGAMP (STING agonist)	LiSmore (This study)	Notes
Switches and tissue penetration/retention	No switch; No limit for tissue penetration	Blue light-switchable; Ultra-light sensitive; can be activated beneath the skin with 1-4 mW blue light at 470 nm	cGAMP: limited specificity with poor tumor or LNs targeting and retention; LiSmore is compatible with stimulation in deep brain regions or shallow LNs of mice through blue light under low light intensity; Future combination with upconverting nanomaterials can extend the depth of tissue penetration to up to 2-3 cm
Spatiotemporal control	Lack of spatial control	Yes	For LiSmore, localized STING pathway activation can be achieved exclusively at photo-stimulated areas
Reversibility	No	Yes	LiSmore shows rapid formation of clusters within seconds upon exposure to blue light, and these clusters could be disassembled upon light withdrawal (half-time: ON: ~1-2 min; OFF: ~30 min)
Abscopal effect	No	Yes	cGAMP has no appreciable abscopal effect due to its relative short half-life and inability to elicit durable STING activation; LiSmore exhibits noticeable abscopal effect, probably owing to its ability to induce rapid and prolonged STING activation in engineered immune cells
Systemic toxicity in vivo	Strong	Weak	cGAMP induces side effects in liver (increase in ALT/AST) and kidney (increase in urea), accompanied with a high serum level of IL-6 (indicative of systemic inflammation); LiSmore-DCs do not seem to cause overt systemic inflammation
Production and delivery	Simple; requires multiple dosages via systemic or intratumoral injection	Sophisticated procedures that involve viral transduction and ex vivo culture; administered intratumorally or systemically	Long-lasting STING agonists are needed to minimize the dosage and frequency of injection; The translational potential of LiSmore can be fully unleashed following near-future breakthroughs in in vivo transduction of selected immune cells

1.2. “To increase the impact of this study, the authors should demonstrate more superior utility of this design over STING agonists. e.g. systemically deliver the construct, then locally activate signaling using focused beam of blue light? Can they control activation more precisely at tumors or at inner organs where STING agonist is lacking?”

Response: We are extremely grateful to Reviewer 1 for this important comment and valuable suggestions. The suggested experiment for systematic delivery of the construct (genetically-encoded materials) will likely have the similar side effects (including systemic toxicity) as existing STING agonists (small molecules based). Systematic delivery of the construct also has the additional technical difficulty with regard to where to administer and how to deliver the genetic construct specifically into immune cells. Without solving the bottleneck for in vivo transduction of selected immune cell populations, it remains not feasible at the current stage to directly compare the LiSmore tool with STING agonists via systematic delivery. From the translational perspective, the most convenient and practical application involves the use of adoptive cell based therapy (similar to CAR-T cell therapy), in which the construct can be engineered into cell types of interest. As we have shown in the manuscript, we used dendritic cell-based immunomodulatory approach via LiSmore introduction into dendritic cells (LiSmore-DC) to demonstrate the efficacy of the optogenetic tool.

First, to determine if “they control activation more precisely at tumors”, we have followed R1’s suggestion to inoculate tumors at both dorsal flanks of mice (a bilateral model of melanoma; **Figure R2A**, new **Figure S10A**). Following tumor establishment on day 7, the right flank (the primary site) of the mice was exposed to a pulsed blue light beam (470 nm at a power density of 2 mW/cm²; 20 sec ON + 5 min OFF; 18 hours), while the left side (the distal site) was shielded from light using aluminum foil. This experimental setup allowed us to examine the spatial control of LiSmore-expressing DCs (cGAMP as a chemical lacks spatial specificity). We found that increased staining of p-TBK1 and p-IRF3 staining was only detected in the primary site subjected to photostimulation, while the distal site shielded from light displayed no appreciable activation (**Figure R2B-C**, or new **Figure S10B-C**). These new results provide compelling evidence for the precise spatial and temporal control achieved through the implementation of our optogenetic strategy.

Figure R2. Spatial control of LiSmore activation in a syngeneic B16-F10 melanoma model.

(A) B16-F10 melanoma cells (3×10^5) were inoculated (s.c.) on both flanks of mice for bilateral tumor establishment, followed by injection of engineered BMDCs (Control or LiSmore; 5×10^5 cells; i.v.) on day 7. The mice were then subjected to photo-stimulation on the right side of tumor (the primary site; 470 nm; ~ 2 mW/cm²; 20 sec ON + 5 min OFF for 18 hours). The left side (the distal site) was shielded from blue light throughout the experiment (dark).

(B) Flow cytometry analysis on the levels of pTBK1 and pIRF3 immunostaining in CD45.2⁺GFP⁺ BMDCs isolated from the indicated groups.

(C) Qualification of the mean fluorescence intensity (MFI) of pTBK1 and pIRF3 staining in CD45.2⁺GFP⁺ BMDCs. $n = 5$ biologically independent mice (means \pm S.D.). ** $P < 0.01$, **** $P < 0.0001$ (Two-sided unpaired Student's t-test).

Second. for deep tissue penetration and demonstration in inner organs, we will have to combine cellular immunotherapy with deep-tissue-penetrating near infrared light (NIR)-responsive nanomaterials as nano-transducer to emit blue light as we and others have illustrated (PMID: 34697491; 33958793; and 26646180), but this seems to be beyond the scope

of this study by focusing on melanoma on skin surface (readily accessible with the ultra-light-sensitive LiSmore tool). It is our near future goal to test the platform in tumors residing within inner organs. Please also see our response to Comment 1.3, where we added a new Table to illustrate the pros and cons of both strategies (**Table R1**, new **Table S2**).

1.3. “What about systemic toxicity, abscopal effects?”

Response: For systemic toxicity assessment, we collected serum samples for ELISA to compare the levels of alanine transaminase (ALT) and aspartate transaminase (AST), urea and IL-6 by LiSmore-DC or cGAMP treatment as previously described (PMID: 35107989, PMID: 33558734). cGAMP treatment resulted in considerable systemic side effects, as evidenced by elevated levels of alanine transaminase and aspartate transaminase (ALT and AST; indicative of liver function), urea (kidney), and serum IL-6 (**Figure R3**, new **Figure S12**). In contrast, the systemic administration of LiSmore-DCs did not appear to cause a higher degree of side effects when compared to the PBS or control groups (**Figure R3**, new **Figure S12**).

Figure R3. Assessment of systemic toxicity after LiSmore-DC or cGAMP treatments.

Measurements were conducted one day after the indicated final treatments in the bilateral B16-F10 melanoma-bearing mice (Related to **Figure 6**). n=5 biologically independent samples for each group (means ± S.D.). **** $P < 0.0001$ (One-way ANOVA.); ns, not significant.

(A) Quantification of serum concentrations of ALT and AST, which serve as indicators of liver function. (B) Quantification of serum urea levels, indicative of kidney function. (C) ELISA measurements of serum levels of IL-6, a key cytokine for assessing systemic inflammation.

For abscopal effects, we generated bilateral B16-F10 melanoma-bearing mice and examined potential abscopal effect on the distal site (left flank) by exposing the primary site (right flank) to pulsed blue light (**Figure R4A**, new **Figure 6A**). In parallel, we performed similar experiments with three intratumoral injections of either cGAMP or PBS on the primary sites using the same melanoma model. Both blue light illumination and cGAMP administration led to significant tumor-suppressive effects at the primary sites (**Figure R4B-C**, new **Figure 6B-C**). Interestingly, the LiSmore group, but not the control or cGAMP groups, showed a noticeable reduction in tumor sizes at the distal sites (**Figure R4B-C**, new **Figure 6B-C**), implying a strong abscopal effect induced by LiSmore upon light stimulation.

Figure R4. Photo-activated LiSmore suppresses distant tumor growth in a syngeneic bilateral B16-F10 melanoma model.

(A) Overview of the experiment design. 3×10^5 B16-F10 melanoma cells without OVA transduction were injected (s.c.) in the right and left flanks of WT B6 CD45.1 mice. Mice received intratumoral PBS, or cGAMP ($10 \mu\text{g}/\text{mouse}$) on the right side at day 7, 10.5 and 14 after tumor inoculation, or were transferred with BMDCs expressing Control or LiSmore (5×10^5 cells/mouse) at day 7. The right side of the tumor (the primary site) was subjected to pulsed blue light stimulation for 7 days (470 nm; $\sim 2 \text{ mW}/\text{cm}^2$; 20 sec ON + 5 min OFF; 6 hr per day), while the left side (the distal site) was shielded from blue light (dark). (B) Left, schematic showing the treatment in the bilateral melanoma model. Right, individual growth curves of tumors in the primary (upper panels) and distant sites (lower panels) for the indicated treatment groups ($n = 5$ mice). (C) Primary and distal tumor growth in the indicated

groups of mice (n=5 per group; mean ± S.D.). **P < 0.01; ***P < 0.001 (Two-sided unpaired Student's t-test).

To compare the antitumor cytokine induction by LiSmore and cGAMP, we analyzed IFN-β and IFN-γ expression at 4 hr and 18 hr after the final treatment (**Figure R5A**, new **Figure S11A**). No significant difference was observed between the LiSmore and cGAMP groups at 4 hours post-treatment. However, at the later time point (18 hours), the LiSmore group displayed higher and more sustained production of IFN-β and IFN-γ in both the primary sites (**Figure R5B**, new **Figure S11B**) and in the sera (**Figure R5C**, new **Figure S11C**) compared to the cGAMP group. The difference might partially explain the observed discrepancy in abscopal effect.

Figure R5. Photo-activated LiSmore causes more sustained production of IFNβ and IFNγ.

(A) Schematic illustration of the experimental timeline and setup. Both flanks were inoculated with melanoma cells and subjected to either photostimulation or cGAMP treatment in the right flank.

(B) Quantification of IFN-β and IFN-γ production in the primary tumor sites (top panels) and sera (bottom panels) obtained from B16-F10 dual tumor-bearing mice at 4 and 18 hr after the final treatment. n=5 biologically independent mouse for each group (mean ± S.D.). **P < 0.01; ***P < 0.001; **** P < 0.0001 (One-way ANOVA); ns, not significant.

Finally, we admit that more follow-on studies are needed to fully understand the potential of optogenetic tools for anti-tumor therapy and mechanism underlying the abscopal effects. We would like to point out that it is difficult to directly compare the two therapeutic modalities given that one is adoptive cell therapy and the other is chemotherapy. In terms of spatial and temporal control over anti-tumor immunity and mitigation of side effects, LiSmore-DC is apparently superior over STING agonists. However, cell-based immunomodulatory therapy requires more procedures and might cost more than STING agonists. In addition to translational studies, the tools described in the manuscript can be applied as a research tool to precisely interrogate signaling events in the cGAS-STING pathway and facilitate future mechanistic studies. We have added a new Table to illustrate the pros and cons of both strategies (**Table R1**, new **Table S2**).

Reviewer 2 (expert in cancer therapy, PD1 engineering):

The manuscript entitled “Optogenetic engineering of STING signaling to remotely control anti-tumor immunity” submitted by Yaling Dou et al., describes the novel noninvasive approach for the development of cancer therapeutics. Investigators manipulate the cGAS-STING signaling pathway by using ultra-light-sensitive optogenetic LiSmore for light-inducible by remotely control STING activation and downstream gene expression in antigen-presenting cells and to activate innate immunity. This well-designed study was meticulously carried out to tackle every step of the process, primer design, cellular studies, transformation, in vitro, ex vivo and in vivo mouse studies.

Investigators systematically start this study from STING signal pathway in the cell (in vitro) and test the hypothesis in pre-clinical models with the transgenic tumor xenografted mice. This is a good example for translational research. The study design, manuscript presentation, data analysis, results and discussion and conclusion were well done. Except minor typos and edits. The depth and breadth of “noninvasive activation of LiSmore” story is quite intriguing, and I feel this manuscript could be well deserved to be published in Nature communications.

Overall, the study results, data validity and significance of the research are good quality based on this assessment **I would like to recommend that this manuscript can be accepted for publication** after addressing the following comments and feedback.

Response to R2 comments

We would like to express our sincere thanks to the reviewer the supportive remark that “*I would like to **recommend that this manuscript can be accepted for publication** after addressing the following comments and feedback.*”

2.1 “*Although this manuscript provided all details regarding data (including SI and video) with smooth flow from figures 1-5 in a sequential manner, but it lacks eagerness to read due to overwhelming acronyms. Authors could have presented a concept figure in such a way to capture the quick overview of the entire study that may be helpful to readers, since abstract do not reflect the entire volume of work performed by the authors.*”

Response: We thank the reviewer for the great suggestion. We have followed the valuable advice to reduce the use of acronyms throughout the revised manuscript. In addition, we have added one concept figure to capture the overall idea and emphasized the motivation of the study in the introduction and discussion sections (see **Figure R1** above, or new **Figure 7**). Please also see our response to Comment 1.1.

2.2. “*The primers list can be moved to SI.*”

Response: We have moved the primers list to SI (new **Table S1**).

2.3. “*Source of some of the mouse strain was not provided e.g., C57BL/6.*”

Response: We have provided all the mouse strain names and the sources for animals used in the study. “C57BL/6-Tg (TcraTcrb) 1100Mjb/J (CD45.2, H-2b) (OT-I) mice, C57BL/6-CD45.1 (B6 CD45.1), and C57BL/6J (B6 CD45.2) mice were purchased from the Jackson laboratory (Bar Harbor, ME, USA).” All mice were maintained in the vivarium according to IACUC approved protocols.

2.4. “*Figure 1E, x-axis list of acronyms needs to be expanded.*”

Response: We have followed the reviewer’s advice to provide further explanations on the acronyms. We have expanded and described x-axis both in the text and revised Figure 1E legend.

Revised text: “We found that blue light illumination induced a sharp increase (8-40 fold) of the STING pathway-related signature genes, including the radical S-adenosyl methionine domain containing 2 (*RSAD2*), C-X-C motif chemokine ligand 10 (*CXCL10*), interferon beta 1 (*IFNB*),

interferon induced protein with tetratricopeptide repeats 1/2/3 (*IFIT1*, *IFIT2*, *IFIT3*), and interferon-stimulated 15 KDa protein (*ISG15*) (**Figure 1E**).”

Figure 1 E legend: “**Figure 1(E)** Quantification of mRNA expression levels of the STING pathway-related signature genes, including the radical S-adenosyl methionine domain containing 2 (*RSAD2*), C-X-C motif chemokine ligand 10 (*CXCL10*), interferon beta 1 (*IFNB*), interferon induced protein with tetratricopeptide repeats 1/2/3 (*IFIT1*, *IFIT2*, *IFIT3*) and interferon-stimulated 15 KDa protein (*ISG15*) with qRT-PCR.”

2.5. “Light-inducible SMOC-like repeats, *CRY2clust-pLxIs*, was abbreviated as *LiSmore*, this could be printed in abstract and one more time at the beginning of the text, but this abbreviation was repeated almost in all instances both in text and figure in parenthesis why?”

Response: In our original manuscript, we repeatedly presented the full construct elements/names for *LiSmore* to better inform the readers about the configuration of the construct. We have followed the reviewer’s advice to add *LiSmore* in the abstract, as well as at its first appearance in the main text, but avoided repeating in the follow-on text and figures/figure legends (**Figure R6**, revised **Figure 2A**). In the revised manuscript and figures, we have referred to the constructs as Control and *LiSmore*, with GFP-*CRY2clust* representing the former and GFP-*CRY2clust-pLxIs* representing the later.

Figure R6. Scheme for the experimental setup. Bone marrows from C57BL/6J mice were cultured in GM-CSF to induce BMDCs. On days 7 and 8, BMDCs were transduced twice with Control (GFP-*CRY2clust*) or *LiSmore* (GFP-*CRY2clust-pLxIs*). After 36 hr, GFP⁺ BMDCs were sorted and replanted in 48-well plates. BMDCs were either shielded from light (Dark) or exposed to blue light (Light; 470 nm, 1 mW/cm²; 20 s ON, 5 min OFF cycles).

2.6. “Figure 3F, in legend it was indicated Left, schematic showing the in vitro coculture system, this was missing only two sections were presented, this need to be corrected.”

Response: We thank the review for pointing out this issue. We have corrected it in our revised manuscript by deleting “schematic showing the in vitro coculture system” in **Figure 3F**. Instead, we revised as follows: “**Figure 3. (F)** Left, representative FACS profiles indicative of IFN- γ production from OT-1 CD8⁺ T cells 18 hours after co-culture with BMDCs/OVAp under the

indicated treatments. Right, quantification of IFN- γ production in OT-1 CD8⁺ T cells. n = 6 independent biological replicates (mean \pm S.D.); **P < 0.01; ***P < 0.001 (one-way ANOVA test).

2.7. *“Figure 4, presentation of data was well laid. Figure 4-D, it looks displaying tumor volume shrinkage was almost equal to cGAMP and LiSmore treated, this was not explained well in main text.”*

Response: We thank the review for pointing out this issue. We have added more sentences in the Discussion to explain this observation. “Our investigations have demonstrated that LiSmore leads to more rapid and sustained activation of downstream effectors, including TBK1 and IRF3 (**Figure 2C**, and **Figure S11**). This prolonged activation may explain the disparities observed in CCR7, MHC-I, and T-cell proliferation in vitro (**Figure 3 C**), as well as the improved antigen presentation and CD80 expression within DCs in vivo (**Figure 4E-F**), between the LiSmore and cGAMP treatment groups. STING activation has been shown to be crucial for the generation of stem-like central memory CD8⁺ T cells (66). Our LiSmore tool may promote T cell memory by employing pulses of STING activation, whereas the continuous activation of STING by cGAMP may lead to T cell death due to over-activation (21, 67). Although LiSmore and cGAMP treatments resulted in comparable levels of tumor clearance in one of our tumor models (**Figure 4D**), LiSmore demonstrated abscopal anti-tumor efficacy in the bilateral tumor model (**Figure 6**). Further studies using tumor challenge models or metastasis models may provide deeper insights into the differential effectiveness of LiSmore and cGAMP in promoting anti-tumor immunity. These models can help to evaluate the ability of the treatments to prevent tumor growth or metastasis, as well as their long-term effects on T cell memory and overall survival.”

Please also see our responses to Comments 3.6 and 3.7 below.

2.8. *“Figure 5C-D, legend marks were common for both sections, it should be laid in center or provide separate for 5D.”*

Response: We have separately added the marks for the revised **Figure 5D (Figure R7B)** as shown in the revised **Figure 5C (Figure R7A)**.

Figure R7. LiSmore enhances anti-PD-L1 treatment efficacy in an immunosuppressive LL/2 lung carcinoma model. (A) LL/2 tumor volumes in the indicated groups of mice (n = 4 per group). (B) Quantification of tumor weights upon sacrifice at day 18. n = 4 mice (mean ± S.D.). *P < 0.05; **P < 0.01 (one-way ANOVA test).

2.9. “SI, Figures S2, and S4, axis or legend labels fonts are printed not clear it is cramped, please fix it.”

Response: We apologize for the mistake when transforming figures to the PDF version using a Mac computer (due to its incompatibility with GraphPad Prism software). This issue was solved by using window PC to convert the graph plotted by the Prism/GraphPad software.

2.10 “Minor typos: Author contributions: It is printed as “date analysis”, it should be fixed.”

Response: We thank the reviewer for pointing out this typo. We have corrected “date” to “data” in the author contribution section.

Reviewer 3 (expert in optogenetics):

The authors developed an optogenetic strategy to activate STING signaling based on inducible clustering of CRY2. They also demonstrated the usage of an ultra sensitive version of CRY2 to reduce light dosage during in vivo treatments. Then they further verified the activation of downstream genes in the STING pathway after light induction, including antigen presenting capability and cytokine producing efficiency. Using the LiSmore strategy, they obtained similar tumor toxicity comparable to cGAMP treatment both ex vivo and in vivo. Finally, combining with the checkpoint immunotherapy, they improved the therapy for tumors with immune-suppressive microenvironment.

The tool the authors developed has been proposed as an alternative for cGAMP treatment with better spatio-temporal and cell-type specificity, which is promising in future therapies. The

development of the protein tools and its successful application in vivo is exciting.

Congratulations to the authors on impressive work. The data are convincing. We only suggest a few minor changes to improve the manuscript.

Response to R3 comments

We thank the reviewer for the enthusiastic and supportive remarks that “*The development of the protein tools and its successful application in vivo is exciting. **Congratulations to the authors on impressive work.** The data are convincing. We **only suggest a few minor changes** to improve the manuscript.*”

3.1. “*The manuscript would be strengthened by more clearly establishing the existing gap that this paper solves, perhaps even in the introduction. The deficiency of cGAMP treatment is only mentioned in the discussion, but would help frame the motivation of the paper.*”

Response: We have revised both the introduction sections to provide the background information regarding the existing gap and to better frame the motivation of the manuscript. Please also see our responses to Comments 1.1 and 2.1 above (see **Figure R1** above, or new **Figure 7**), as well as to Comment 1.3. (See **Table R1** above, or new **Table S2**) summarizing LiSmore versus STING agonist (cGAMP as the most studied example) in anti-tumor immunity.

For instance, in the introduction we added the following sentences:

“However, the use of STING agonists raises a significant concern regarding the potential induction of systemic inflammation due to their pleiotropic effects on various immune cell types. Furthermore, achieving effective concentration and retention within tumors often necessitates high dosages, which can exacerbate side effects (9). Conversely, genetically engineering STING modulators within specific immune cell populations holds promise for enhancing spatiotemporal control and cell-type specificity, thus minimizing side effects associated with STING agonist treatment.”

“Contrary to its recognized tumor-suppressive role, recent studies have suggested the involvement of the STING pathway in promoting tumor burden and contributing to poorer disease outcomes in murine tumor models (17, 18). While transient activation of the pathway appears to favor tumor suppression, the lasting STING activation can result in chronic inflammation, creating an immunosuppressive tumor environment to promote tumor growth (2, 19). In parallel, STING agonists have the potential to induce the expression of inhibitory molecules, such as programmed death-ligand 1 (PDL1) and indoleamine 2, 3-dioxygenase 1 (IDO1), which counteract the tumor-suppressive effects (2, 20). Moreover, excessive activation

of the STING pathway has been shown to promote apoptosis in T and B cells (21, 22). Additionally, as STING is expressed ubiquitously in multiple cell types, the administration of STING agonists may lead to systemic inflammation, including the development of inflammatory and autoimmune responses (2). These findings underscore the critical need to develop a method that would allow precise temporal and spatial control over the STING signaling, particularly in antigen-presenting cells such as dendritic cells, in order to mitigate the aforementioned adverse effects.”

3.2. *“Along the same lines, the authors should make a good case for why optogenetics is important for this use case. Given that perturbations are not done in a fast timescale, and that light control would be limited only to those situations where light could be administered topically, it seems that a chemically-induced protein aggregation approach might be more effective (drug is applied systemically but only acts on the engineered DCs). What benefit does optogenetics offer here relative to other control modes?”*

Response: Chemically-induced protein aggregation approach often involves the use of rapamycin to induce FKBP-FRB multimerization. Such chemically-induced dimerization (CID) system lacks strict spatial precision. Additionally, the rapamycin/rapalog chemical induced dimerization system is limited by its quasi-irreversibility (PMID: 36163385, PMID: 20353181). Furthermore, rapamycin is used as an immunosuppressant in the clinic, which might counter the immunostimulatory effect following STING activation. By contrast, we observed reversible clustering of LiSmore and repeated recruitment of TBK1 in response to ON/OFF cycle of blue light stimulation (**Figure R8**, new **Figure S3** and new **Supplementary Movie 3**). Therefore, the LiSmore tool could circumvent these caveats by conferring both spatial and temporal and control over the STING pathway in a reversible manner.

We add description in the introduction as “Chemically-inducible dimerization (CID) have been developed as a chemogenetic approach for tunable control of protein oligomerization. However, CID systems lack strict spatial precision and often suffer from irreversibility (23). In contrast, optogenetics offers tremendous potential for achieving precise spatial and temporal control over physiological processes in live cells and tissues (24-28).”

Figure R8. Reversible co-clustering of LiSmore with TBK1 in response to two ON/OFF cycles of photo-stimulation. HEK293T cells co-expressing mCherry-LiSmore (red) and TBK1-YFP (green) were stimulated by the built-in 488-nm laser source (5% input). Time-lapse confocal imaging showed the reversible clustering of LiSmore with subsequent recruitment of TBK1 in response to two ON/OFF cycles of photostimulation. Scale bar, 20 μ m. Also see **Supplementary Movie 3**.

3.3. Given phototoxic effects of blue light, the authors should include a negative control in 1E, for example cells transfected with Cry2-Ctrl (as in 1F).

Response: We sincerely thank the reviewer for the control suggestion. We have added CRY2-Ctrl-expressing HEK293T cells as the negative control (omitted in the original manuscript) in the revised **Figure 1E (Figure R9)**. Because we used pulsed blue light with rather low power densities (0.1-4 mW/cm²), we minimized potential phototoxicity or unspecific effects induced by light pulses during our experimental window.

Figure R9. Quantification of mRNA expression levels of the STING pathway-related signature genes. The mRNA expression level of radical S-adenosyl methionine domain containing 2 (*RSAD2*), C-X-C motif chemokine ligand 10 (*CXCL10*), interferon beta 1 (*IFNB*), interferon induced protein with tetratricopeptide repeats 1/2/3 (*IFIT1*, *IFIT2*, *IFIT3*) and interferon-stimulated 15 KDa protein (*ISG15*) were detected by qRT-PCR.

3.4. “The comparisons between Cry2 variants in Fig1G are only interpretable when the expression levels of the tools are the same. Were these cell lines expression matched?”

Response: We are really grateful to the reviewer for this important comment. These cell lines were matched with their overall mCherry expression levels. We have sorted the transfected HEK293T cells for mCherry positive cells and detected the expression of comparable mCherry signals by flow cytometry for verification. Please see below for representative FACS profiles:

Figure R10. mCherry expression was quantified by FACS after sorting. HEK293T cells expressing the indicated constructs were sorted based on mCherry expression and quantified by FACS.

We added the related description in the revised Method: “For some experiments, HEK293T cells expressing the indicated constructs were first sorted with flow cytometry based on mCherry fluorescence signals, and then seeded with matched mCherry expression (>90%) for gene expression and IFN- β detection.”

In the revised figure legend, “Figure 1. (G) Determination of secreted IFN β using ELISA at varying light intensities. FACS-sorted HEK293T cells expressing the indicated constructs (with comparable mCherry) were either kept in the dark or exposed to pulsed blue light delivered at 0.1, 0.5, 1 and 2 mW/cm² (30 s ON and 30 s OFF for 8 hours). n = 4 independent biological replicates (mean \pm S.D.); *P < 0.05; **P < 0.01; ****P < 0.0001 (Two-sided unpaired Student’s t-test).

3.5. “Since 3I is a coculture of 3 cell types, can the authors use a more specific assay to verify death of the B16-OVA? It seems LDH release only informs that some cells are dying, not which ones.”

Response: We thank the reviewer for bring this potential caveat to our attention. We have followed the reviewer’s great advice to perform additional flow cytometry-based experiments to verify specific cell death in the target cells. Before co-culturing B16-OVA cells with LiSmore-transduced BMDCs and CD8⁺ T cells, we pre-stained the B16-OVA cells with CellTrace Violet

dye. This was done to discriminate B16-OVA tumor cells from BMDCs or CD8⁺ T cells, which were Violet-negative. The results of these experiments were presented in the new **Figure S6 (Figure R11)**. After a 24-hour co-culture period, we collected the cells and stained them with 7-AAD, a fluorescent intercalator commonly used to identify non-viable cells. We then analyzed the cell population using flow cytometry and identified the death of the pre-stained B16-OVA cells by gating on the Violet⁺7-AAD⁺ double-positive population (**Figure R11B-C**, new **Figure S6B-C**). The following text has been added to the revised manuscript:

“To validate the specificity of T cell-mediated cytotoxicity toward tumor cells, B16-OVA melanoma cells were pre-stained with CellTrace Violet prior to co-culturing with BMDCs and CD8⁺ T cells. Flow cytometry analysis was then conducted to assess B16-OVA cell death, indicated by double-positive staining of the violet dye and 7-AAD (**Figure S6A**). Upon blue light illumination, the LiSmore group demonstrated enhanced T cell-mediated cytotoxicity towards B16-OVA cells. The extent of B16-OVA cell death (Violet⁺7-AAD⁺) was approximately 2-4 fold higher in the LiSmore group upon photostimulation compared to either the cGAMP group or the control groups (**Figure S6B-C**). These compelling findings establish that LiSmore efficiently promotes the activation of antigen-presenting cells and enhances cross-presentation of tumor antigens, effectively priming effector CD8⁺ T cells for targeted tumor killing in a light-dependent manner.”

Figure R11. Flow cytometry analysis on CD8⁺ T cell-mediated cytotoxicity towards B16-OVA melanoma cells. (A) Schematic showing the in vitro co-culture system. OT-1 CD8⁺ T cells were incubated with a mixture of LiSmore-BMDCs (pre-stained by CellTrace Violet; Violet⁺) and B16-OVA cells at a 2:1:1 ratio with or without light illumination (470 nm, 1 mW/cm²; 20 s ON, 5 min OFF; 24 hours). 7-AAD staining was utilized to detect cell death. (B) Representative FACS

profiles for the indicated groups that report B16-OVA melanoma cell death (Violet⁺7-AAD⁺) 24 hours after co-culture. (C) Quantification of T cell-mediated cytotoxicity toward B16-OVA cells as indicated by the percentage of dead B16-OVA cells (Violet⁺7-AAD⁺). n=3 independent biological replicates (mean ± S.D.). **P < 0.01; **** P < 0.0001 (Two-sided unpaired Student's t-test).

3.6. “Since the authors use cGAMP stimulation as a positive control throughout the paper, it would be helpful to see the effect of cGAMP on pTBK and pIRF3, for example in figure 2. Could differences in pTBK and pIRF3 between LiSmore and cGAMP explain later differences in CCR7 level, MHC-1 level and T-cell proliferation?”

Response: We thank the reviewer for your suggestion regarding how the effect of LiSmore and cGAMP differs on pTBK and pIRF3. We agree that comparing the effect of LiSmore and cGAMP on these proteins would provide valuable insights and enhance the comprehensiveness of our study. We have followed the reviewer's suggestion to compare the effect of cGAMP and LiSmore on p-TBK and p-IRF3 by western blot side-by-side. Interestingly, we observed differential activation patterns between LiSmore and cGAMP treatment. In the cGAMP-treated group, we noticed a relatively delayed onset of TBK1/IRF3/P65 phosphorylation, which peaked around 4 hours after cGAMP treatment, followed by a rapid decline at 6 and 8 hr post-treatment, indicating a transient and time-limited activation of the STING pathway in response to cGAMP stimulation (**Figure R12**, new revised **Figure 2C**). In contrast, LiSmore showed an earlier and more sustained activation profile of the STING pathway. The phosphorylation of TBK1/IRF3/P65 was observed within 1 hour post-photostimulation and persisted for the entire 8-hour duration (**Figure R12**, new revised **Figure 2C**). These findings highlight the unique properties of LiSmore as an engineered innate immunity actuator, capable of eliciting sustainable and prolonged activation of the innate immune response through the STING pathway. The observed differences in CCR7 level, MHC-1 level, and T-cell proliferation could potentially be attributed to the variations in pTBK and pIRF3 between LiSmore and cGAMP treatments, as well as the distinct activation profiles.

Figure R12. Differential activation patterns between LiSmore and cGAMP treatment.

Immunoblot analysis of TBK1, IRF3, and p65 phosphorylation in BMDCs in the absence (black bar) or presence of light stimulation (blue bar; 470 nm, 1 mW/cm²; 20 s ON, 5 min OFF cycles) for the indicated durations (0-8 hr). Non-transduced BMDCs were treated with 2'3'-cGAMP (2 µg/ml) under the same timeframe side-by-side. Results are representative of two independent experiments.

We added the following sentences in the main text to reflect the new data: “To gain a mechanistic understanding of LiSmore-mediated effect on dendritic cells, we further examined the TBK1/IRF3/P65 phosphorylation levels in LiSmore-transduced BMDCs by immunoblotting. For strict comparison, we included CRY2clust as a negative control and cGAMP treatment as a positive control. Interestingly, we observed distinct activation profiles of the downstream effectors within the STING pathway when comparing LiSmore and cGAMP treatment. In the cGAMP-treated group, we noticed a relatively delayed onset of TBK1/IRF3/P65 phosphorylation around 4 hours after cGAMP treatment. Moreover, this phosphorylation substantially declined at 6 and/or 8 hours post-treatment, indicating a transient and time-limited activation of the STING pathway in response to cGAMP stimulation. In contrast, LiSmore showed an earlier and more sustained STING activation profile, appearing within 1 hour post-photostimulation and persisting over the course of 8 hours (**Figure 2C**). Collectively, these findings demonstrate that LiSmore, as an engineered innate immunity actuator, is capable of eliciting sustainable and prolonged activation of innate immune response through the STING pathway. This further highlights the unique and advantageous properties of LiSmore in modulating innate immunity for potential therapeutic intervention.”

In the discussion section, we further added the following sentences: “Our investigations have demonstrated that LiSmore leads to more rapid and sustained activation of downstream effectors, including TBK1 and IRF3 (**Figure 2C**, and **Figure S11**). This prolonged activation may explain the disparities observed in CCR7, MHC-I, and T-cell proliferation in vitro (**Figure 3 C**), as well as the improved antigen presentation and CD80 expression within DCs in vivo (**Figure 4E-F**), between the LiSmore and cGAMP treatment groups. STING activation has been shown to be crucial for the generation of stem-like central memory CD8⁺ T cells (66). Our LiSmore tool may promote T cell memory by employing pulses of STING activation, whereas the continuous activation of STING by cGAMP may lead to T cell death due to over-activation (21, 67). Although LiSmore and cGAMP treatments resulted in comparable levels of tumor clearance in one of our tumor models (**Figure 4D**), LiSmore demonstrated abscopal anti-tumor efficacy in the bilateral tumor model (**Figure 6**). Further studies using tumor challenge models or metastasis models may provide deeper insights into the differential effectiveness of LiSmore and cGAMP in promoting anti-tumor immunity. These models can help to evaluate the ability of the treatments to prevent tumor growth or metastasis, as well as their long-term effects on T cell memory and overall survival.”

3.7. *“The authors should discuss why LiSmore and cGAMP might give equivalent levels of tumor clearance despite better antigen presentation capacity and higher CD80 of the LiSmore DCs compared to cGAMP. Are there other pathways controlled by STING that might drive the differences?”*

Response: We reason that LiSmore is specifically activated in dendritic cells to boost higher CD80 expression and promote antigen presentation, while cGAMP did not specifically engage dendritic cells. With regard to the ultimate treatment outcomes, we posit that multiple factors might account for the equivalent level of tumor clearance. We have added sentences in the discussion section to reflect these possibilities (also see our response to Comments 3.6 above).

One possibility is that there are other pathways controlled by STING that might be responsible for driving the difference. For example, in a recent paper published in *Science* (PMID: 36795805), CD5⁺ dendritic cells are particularly critical for effective anti-tumor immune response. It is possible that LiSmore might preferentially activate such specific subset of DCs while cGAMP might activate a broader range of DC subsets that have similar potent anti-tumor activity.

Another possibility is that there may be other pathways controlled by STING that are not affected by the differences in antigen presentation capacity and CD80 expression. These factors are important for optimal T cell activation, while other immune cells and factors may also play important roles in tumor clearance that are not directly related to augmented antigen

presentation and CD80 expression. For example, natural killer cells and macrophages, along with cytokines produced by them, can also contribute to anti-tumor immunity and may be activated differently by cGAMP and LiSmore treatments.

Third, LiSmore-DC and cGAMP induces differential STING activation with varying kinetics or duration of STING activation, hence leading to differential effects on downstream pathways and ultimately affecting tumor recession. It has been demonstrated that STING activation was essential for the generation of stem-like central memory CD8⁺ T cells (PMID: 32581136). It is possible that our LiSmore tool can promote T cell memory by employing pulses of STING activation, whereas the continuous activation of STING by cGAMP may lead to T cell death due to over-activation (PMID: 28484079). Further studies using tumor challenge models or metastasis models can provide more insights into the effectiveness of LiSmore and cGAMP in promoting anti-tumor immunity. These models can help to evaluate the ability of the treatments to prevent tumor growth or metastasis, as well as the long-term effects on T cell memory and overall survival.

Overall, more follow-on studies (but beyond the scope of this technology-oriented paper) are needed to fully understand the mechanisms underlying the therapeutic outcomes. Regardless of these complications, it is encouraging to observe effective therapeutic outcomes in rodent models, particularly in tumors that are non-responsive to anti-PDL-L1 treatment by using LiSmore, as well as the impressive abscopal effect exerted by LiSmore (but not for cGAMP).

3.8. *“For Experiments in Figure 2 and Figure 3, can the authors comment on why they transduce cells on consecutive days? Is this necessary?”*

Response: In our pilot study, we compared the percentage of GFP positive cells transduced once or twice after collecting them on the same day (Day 10). It turned out that the GFP expression level increased from 30% to 55% upon a second transduction. Therefore, in order to obtain more GFP⁺ LiSmore-expressing BMDCs for the following adoptive transfer experiments, we transduced BMDCs twice on consecutive days. We have added a sentence to explain this in the revised Method. “To increase the transduction efficiency in some experiments, we repeated the transduction on consecutive days (73).”

3.9. *“The text in the supplementary figures often rendered incorrectly.”*

Response: We apologize for the confusion caused when transforming figures/texts to the PDF version using a Mac computer. This issue was solved by using window PC to convert the graph plotted by the Graphpad/Prism package.

REVIEWERS' COMMENTS

Reviewer #1 (expert in cGAS/STING, STING signalling):

The authors have addressed most of my concerns. The revised manuscript is much improved and represents a novel design of STING activator that will have unique therapeutic potential.

Reviewer #2 (expert in cancer therapy, PD1 engineering):

The revised (R1) manuscript titled "Optogenetic engineering of STING signaling to remotely control anti-tumor immunity" submitted by Yaling Dou et al., now looks fine. Quality of the presentation improved considerably. The figures are legible to read, appreciate the authors for the through work, in addition to addressing all the comments raised by the reviewers. I am very delighted to recommend for the acceptance of the revised manuscript for the publication. Congratulations for the authors for the excellent work.

Reviewer #3 (expert in optogenetics):

The authors addressed all of my concerns.

Responses to comments

Reviewer #1 (expert in cGAS/STING, STING signalling):

The authors have addressed most of my concerns. The revised manuscript is much improved and represents a novel design of STING activator that will have unique therapeutic potential.

Reviewer #2 (expert in cancer therapy, PD1 engineering):

The revised (R1) manuscript titled “Optogenetic engineering of STING signaling to remotely control anti-tumor immunity” submitted by Yaling Dou et al., now looks fine. Quality of the presentation improved considerably. The figures are legible to read, appreciate the authors for the through work, in addition to addressing all the comments raised by the reviewers. I am very delighted to recommend for the acceptance of the revised manuscript for the publication. Congratulations for the authors for the excellent work.

Reviewer #3 (expert in optogenetics):

The authors addressed all of my concerns.

Response: We would like to thank all three reviewers for their supportive remarks and strong recommendations to accept the revised manuscript.